# Knockout or inhibition of USP30 protects dopaminergic neurons in a Parkinson's disease mouse model

Tracy-Shi Zhang Fang[1] ✉, Yu Sun [2], Andrew C. Pearce[3], Simona Eleuteri [1], Mark Kemp[3], Christopher A. Luckhurst[3], Rachel Williams[3], Ross Mills[3], Sarah Almond[3], Laura Burzynski [3], Nóra M. Márkus [3], Christopher J. Lelliott[4], Natasha A. Karp[4], David J. Adams [4], Stephen P. Jackson [3,5,6], Jin-Feng Zhao[7], Ian G. Ganley [7], Paul W. Thompson[3] ✉, Gabriel Balmus [2,8] ✉ & David K. Simon [1]

Mutations in SNCA, the gene encoding α-synuclein (αSyn), cause familial Parkinson's disease (PD) and aberrant αSyn is a key pathological hallmark of idiopathic PD. This α-synucleinopathy leads to mitochondrial dysfunction, which may drive dopaminergic neurodegeneration. PARKIN and PINK1, mutated in autosomal recessive PD, regulate the preferential autophagic clearance of dysfunctional mitochondria ("mitophagy") by inducing ubiquitylation of mitochondrial proteins, a process counteracted by deubiquitylation via USP30. Here we show that loss of USP30 in *Usp30* knockout mice protects against behavioral deficits and leads to increased mitophagy, decreased phospho-S129 αSyn, and attenuation of SN dopaminergic neuronal loss induced by αSyn. These observations were recapitulated with a potent, selective, brain-penetrant USP30 inhibitor, MTX115325, with good drug-like properties. These data strongly support further study of USP30 inhibition as a potential disease-modifying therapy for PD.

A large body of evidence implicates dysfunction of mitochondrial homeostasis as a key pathophysiological mechanism in Parkinson's disease (PD)[1–8]. Thus, maintenance of a healthy pool of functioning mitochondria requires a system for selectively degrading dysfunctional mitochondria ("mitophagy")[9]. Autosomal recessive PD (AR-PD) due to PARKIN deficiency[10] links PD directly to a defect in mitophagy. In response to mitochondrial dysfunction, PARKIN translocates to the outer mitochondrial membrane where it interacts with PINK1 (another gene where mutations cause AR-PD[11,12]) to ubiquitylate mitochondrial

proteins, thereby inducing fusion of mitochondria with autophagosomes, followed by autophagic degradation[13,14].

We hypothesize that defective mitophagy may also exacerbate α-synuclein (αSyn) toxicity. Point mutations in the αSyn gene[15–20], or duplications or triplications in the gene[21–23], cause autosomal dominant PD (AD-PD). αSyn induces mitochondrial complex I dysfunction, potentially by directly binding to TOM20 on the mitochondrial membrane and thereby interfering with mitochondrial protein import[24]. In addition, dysfunctional mitochondria produce increased

[1]Department of Neurology, Beth Israel Deaconess Medical Center and Harvard Medical School, Boston, MA, USA. [2]UK Dementia Research Institute at the University of Cambridge and Department of Clinical Neurosciences, University of Cambridge, Cambridge Biomedical Campus, Cambridge CB2 0AH, UK. [3]Mission Therapeutics Ltd. Glenn Berge Building, Babraham Research Campus, Cambridge CB22 3FH, UK. [4]Wellcome Sanger Institute, Cambridge CB10 1SA, UK. [5]The Gurdon Institute and Department of Biochemistry, University of Cambridge, Cambridge CB2 1QN, UK. [6]Cancer Research UK Cambridge Institute, University of Cambridge, Cambridge Biomedical Campus, Cambridge CB2 0RE, UK. [7]MRC Protein Phosphorylation and Ubiquitylation Unit, University of Dundee, Dundee DD1 5EH, UK. [8]Department of Molecular Neuroscience, Transylvanian Institute of Neuroscience, 400191 Cluj-Napoca, Romania. ✉e-mail: szhang9@bidmc.harvard.edu; PThompson@missiontherapeutics.com; gb318@cam.ac.uk

reactive oxygen species (ROS), consistent with increased markers of oxidative damage in PD brains, and ROS can increase αSyn accumulation, thus fueling a self-accelerating pathological loop[25–33]. However, the role of mitophagy in clearing away dysfunctional mitochondria in the setting of αSyn induced mitochondrial impairment in vivo is unknown. Indirect evidence for a possible role in this setting comes from findings that PINK1 KO rats show enhanced vulnerability to αSyn toxicity[34].

Most strategies to modulate mitophagy also alter autophagy in general, or impact other steps in the autophagy-lysosome pathway[35], making it difficult to study mitophagy specifically[36,37]. A target that could allow specific molecular manipulation of mitophagy is USP30. USP30 is a deubiquitylating enzyme (DUB) tethered to the outer mitochondrial membrane where it directly removes ubiquitin attached by PARKIN or other E3 ligases[38–40], thereby counteracting PARKIN's ability to promote mitophagy[39,41,42]. As such, siRNA-mediated depletion of USP30 rescues mitophagy in PARKIN-deficient cells and protects dopaminergic (DA) neurons in PARKIN-deficient Drosophila[40,43] and human neurons in cell culture[42,44]. Thus, inhibition of USP30 is an attractive therapeutic strategy for restoring mitophagy to achieve neuroprotection in PD. We now report data demonstrating that disruption of USP30 in *Usp30* KO mice stimulates mitophagy and results in highly significant protection against αSyn toxicity. Further, we report that these effects can be recapitulated by a potent and highly selective brain-penetrant small molecule, MTX115325, with drug-like properties. Together, these data validate USP30 as a potential therapeutic target for neuroprotection in PD.

## Results

### Usp30 KO mice are viable and show no overt pathology
To generate *Usp30* KO in mice, we flanked the *Usp30* essential exon 4 with loxP sites (conditional ready allele) and then deleted it using CRE recombinase (constitutive allele) to generate *Usp30* KO mice (Fig. 1a). The successful deletion of the *Usp30* gene was confirmed by lack of *Usp30* mRNA in the brain, kidney, heart, skeletal muscle, spleen, liver, pancreas and testis (Fig. 1b). Furthermore, no USP30 protein was detected in the cortex of *Usp30* KO mice (Fig. 1c). As previously reported[45], *Usp30* KO mice were born at expected mendelian frequencies (Fig. 1d). In addition, we now perform high-throughput phenotyping of over 300 phenotypic parameters and show that *Usp30* KO mice have no overt pathologies (Supplementary Fig. 1 and Supplementary Data 1). To see if loss of USP30 leads to pathologies with age, we established an ageing cohort and show that USP30 loss has no detectible deleterious effects with ageing when compared to wildtype (WT) littermate controls (Fig. 1e). In fact, we noticed that compared to the WT littermate controls (C57BL/6 N background), 1-year-old *Usp30* KO mice are protected from fatty liver accumulation (Supplementary Fig. 2). Taken together, these data revealed no adverse effects from USP30 loss in mice.

### Knockout of *Usp30* enhances mitophagy levels in dopaminergic neurons
To test the hypothesis that USP30 depletion would affect mitophagy in dopaminergic neurons, we crossbred *Usp30* KO mice with *mito*-QC reporter mice, which have a GFP-mCherry tandem fused to a mitochondrial localization signal derived from the protein FIS1[46]. This makes it possible to measure mitophagy in vivo as the GFP signal is quenched in the acidic environment of lysosomes during mitophagy[46]. Thus, red mCherry puncta without a green GFP signal reflects mitochondria fused with lysosomes during mitophagic degradation (Fig. 1f). Colocalisation of red mCherry puncta with LAMP1 was used as an alternative strategy for measuring mitophagy in brain sections where the endogenous signal of mCherry-GFP is not easily detected with confocal microscopy (Supplementary Fig. 3c).

To understand if USP30 loss can affect mitochondrial clearance, we quantified the mitophagy signals in dopaminergic neurons of *mito*-QC/*Usp30 KO* mice compared to *mito*-QC WT littermates at 16 weeks of age. In this scenario, in individual SN dopaminergic neurons (SNpc; TH-positive, blue; Fig. 1g), we quantified the number of mCherry positive puncta (*mCherry*, red) colocalized with a lysosomal marker (LAMP1, green; Fig. 1f central left panels) representing mitophagosomes fused with lysosomes. We found that *mCherry* puncta are significantly and specifically increased in the dopaminergic neurons of *Usp30* KO mice compared with WT mice (8.8 ± 0.6 per DA neuron in WT mice and 12.7 ± 1.5 per DA neuron in *Usp30* KO mice, $n = 13–14$, $p = 0.0264$; Fig. 1h). We also found the mCherry/mitophagy puncta were not changed in other peripheral tissues such as muscle (Supplementary Fig. 3a, b) but were significantly increased in cortical neurons and hippocampus neurons from *Usp30* KO mice compared with *Usp30* WT mice at 40 weeks of age (Supplementary Fig. 3c, d). Taken together, our results indicate that loss of USP30 in mice increases basal levels of mitophagy in DA neurons in the SNpc, cortical neurons and hippocampal neurons, but not in muscle (Suppl. Fig. 3).

### *Usp30* KO attenuates dopaminergic neuronal loss in the AAV-A53T-SNCA mouse model
To test whether enhancement of mitophagy in DA neurons of *Usp30* KO mice is associated with protection of DA neurons from αSyn toxicity, we used a validated AAV1/2-A53T-SNCA αSyn overexpression PD mouse model that shows dopaminergic neurodegeneration and motor deficits in rat, mouse and non-human primate models[47–51]. Firstly, to determine if induction of αSyn affects mitophagy, we measured mitophagy within the SNpc in ipsilateral (AAV-A53T-SNCA injected) as compared to contralateral sites (Not-injected; NI) at 28 weeks post-delivery. To verify that the *mito*-QC subcellular localization is specific for mitophagy, we also used staining with the mitochondrial marker, OPA-1 and we observed similar results to data generated with the *mito*-QC reporter system (Supplementary Fig. 4a, b). Mitophagy puncta were significantly increased in both the contralateral and ipsilateral SNpc DA neurons of *Usp30* KO mice (15.80 ± 6.837 and 14.29 ± 6.862 per DA neuron, respectively) compared with *Usp30* WT mice (8.889 ± 3.833 and 9.640 ± 4.881 per DA neuron, respectively) after unilateral AAV-A53T-SNCA injection (Supplementary Fig. 4b). Interestingly, expression of mutant αSyn did not affect the basal level of mitophagy independent of USP30 loss, suggesting no direct correlation between accumulation of αSyn and USP30-dependent mitophagy control at the timepoint assessed (Supplementary Fig. 4b). To determine if loss of USP30 protects against αSyn-induced DA neuronal loss in the *Usp30* KO mice, we performed TH+ neuronal counting within the SNpc in ipsilateral (AAV-A53T-SNCA injected) as compared to contralateral (NI) sites at 28 weeks after the injections (Fig. 2a). These results show that USP30 loss significantly attenuated the DA neuronal loss caused by αSyn overexpression (AAV-A53T-SNCA injection; 47.35 ± 4.614 % in WT mice, $p < 0.0001$; 29.47 ± 6.412 % in *mito*-QC mice, $p = 0.0025$; 66.15 ± 3.135 % in *mito*-QC/*Usp30* KO mice, $p < 0.0001$; one sample $t$ test comparing with 100%, Fig. 2b) compared with WT controls ($p = 0.0043$; Fig. 2b) or mito-QC mice ($p = 0.0002$; Fig. 2b). Thus, USP30 absence protects against αSyn-induced DA neuronal loss.

### *Usp30* KO inhibits development of αSyn pathology and associated motor deficits
To determine if upregulation of mitophagy in *Usp30* KO mice injected with AAV-A53T-SNCA is associated with decreased αSyn pathology, we did immunostaining in the SNpc with anti-phospho S129-αSyn, a pathological form of αSyn found in Lewy bodies[52]. Injection of the AAV-empty vector (AAV-Ev) did not produce pathological αSyn in the ipsilateral SNpc of WT, *mito*-QC and *mito*-QC/*Usp30* KO mice,

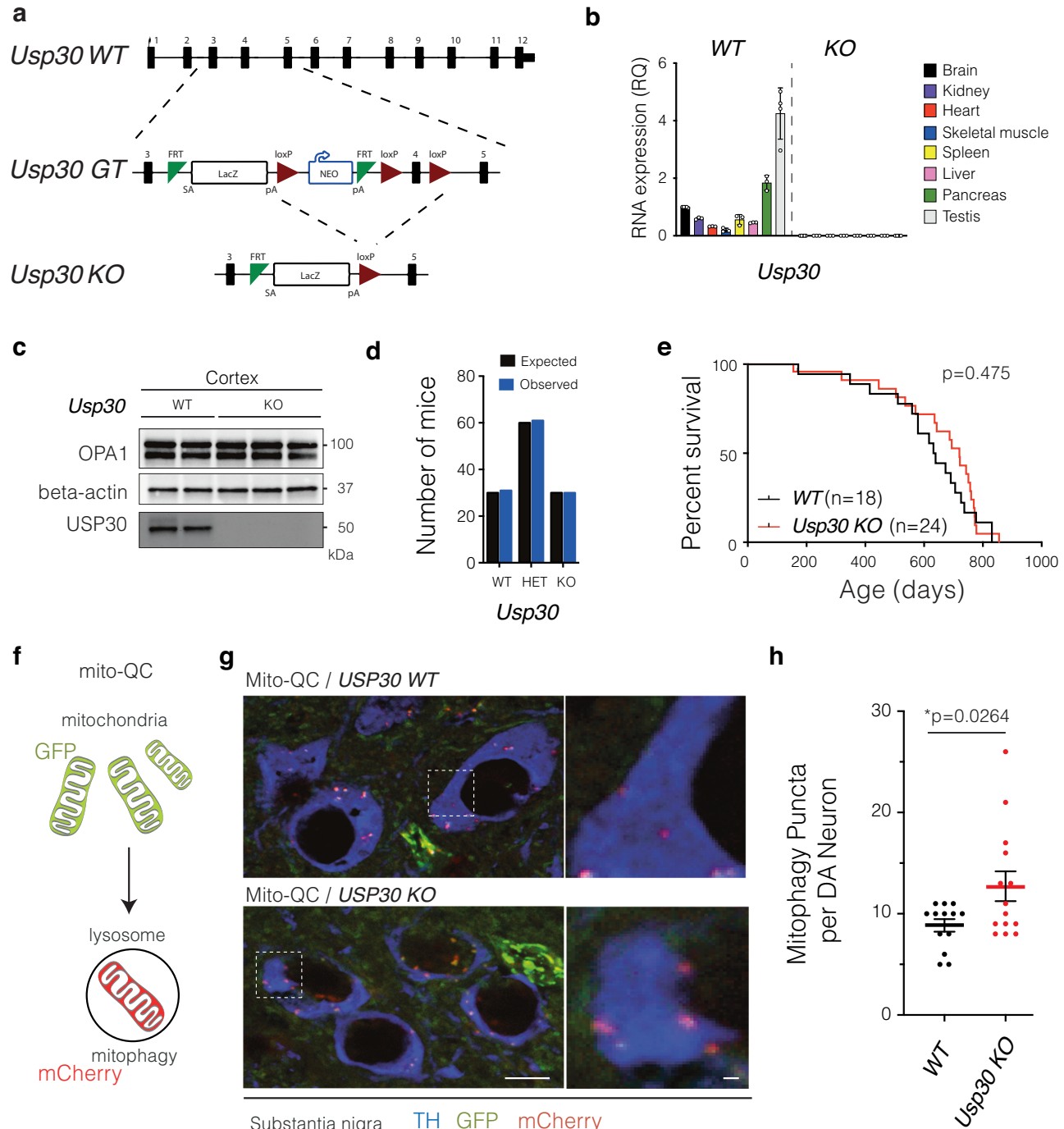

**Fig. 1 | Generation and characterization of *Usp30* KO mice. a** Schematic of gene targeting to generate the *Usp30* KO mice. **b** Bar graph shows the levels of *Usp30* gene expression in different tissues in *Usp30* WT and KO mice (*n* = 5 for brain, *n* = 4 for testis, *n* = 3 for all other tissues). Error bars represent mean ± s.d. **c** Representative Western Blot images of OPA1, beta-actin and USP30 in the cortex of *Usp30* WT and KO male mice. The experiment was repeated twice independently. **d** Estimated and observed numbers of WT, *Usp30* heterozygous (Het) and *Usp30* homozygous knockout (KO) mice in the offspring of heterozygous *Usp30* breeders. **e** Survival curve of WT and *Usp30* KO mice. **f** schematic image showing the working mechanism of *mito*-QC reporter protein for assessing the mitophagy signal in cells. **g** Representative fluorescence images show the *mito*-QC fluorescence signal

(mCherry-red, GFP-green), and dopaminergic neurons (TH, blue) in the SNpc of *mito*-QC and *mito*-QC/*Usp30* KO male mice. Dashed white inlets were enlarged in right panels showing the details of mCherry only puncta (mitophagy puncta) in the DA neurons. Scale bar, 10 μm. **h** Quantification of mitophagy puncta in individual dopaminergic neurons of the SNpc (*n* = 13 for USP WT male mice, *n* = 14 for *Usp30* KO male mice, 5–10 neurons per mouse). WT and *Usp30* KO in the bar graph represent *mito*-QC and *mito*-QC/*Usp30* KO, respectively. Significance determined by unpaired, two-tailed Student's *t* test. Error bars represent mean ± s.d.; *P < 0.05. Acronyms: WT wild type, GT gene-trap, KO knock out, loxP locus of X-over P1 site, FRT flippase recognition target, SA splice acceptor, pA polyA tail, NEO neomycin resistance cassette, LacZ β-galactosidase. Source data are provided as a Source Data file.

respectively (Supplementary Fig. 5a). In contrast, there was robust phospho-S129- αSyn fluorescence immunostaining in the ipsilateral SNpc of AAV-A53T-SNCA injected WT mice (67.10 ± 4.899 in AAV-A53T-SNCA versus AAV-Ev, *p* < 0.0001; Fig. 2c, d) and *mito*-QC mice (76.62 ±

4.854 in *mito*-QC AAV-A53T-SNCA versus AAV-Ev injected mice, *p* < 0.0001; Fig. 2c, d). Notably, the intensity of phospho-S129 αSyn in dopaminergic neurons of the AAV-A53T-SNCA injected mice was significantly reduced in *mito*-QC/*Usp30* KO mice (Fig. 2c, d; 21.09 ± 3.065

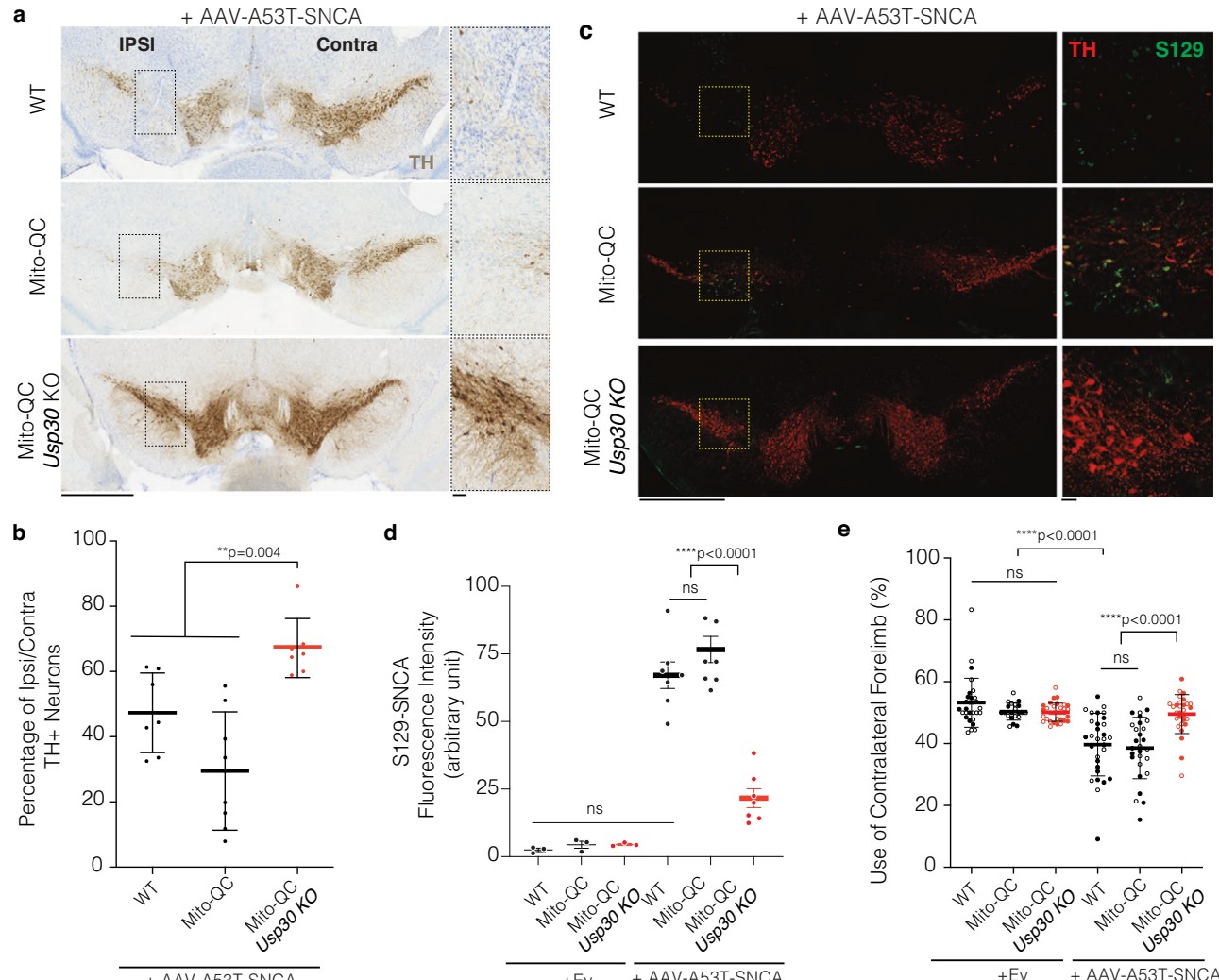

**Fig. 2 | *Usp30* KO improved DN neuronal survival, decreased pathological αSyn and prevented motor deficits in an α-synuclein-induced PD mouse model.**
**a** Representative SNpc sections of TH immunohistochemistry in different groups of male mice at 28 weeks post-injection of AAV-A53T-SNCA. Inset of left panels is enlarged to right panels. Scale bar, 1 mm for left panels, 100 μm for right panels. **b** Graph shows the percentages of TH-positive neurons in ipsilateral compared to contralateral SNpc of the same male mouse in each group ($n = 7$ for WT group, $n = 8$ for *mito*-QC group, $n = 7$ for *mito*-QC/*Usp30* KO group). Significance determined by one-way ANOVA. Error bars represent mean ± s.d.; **$P < 0.01$. **c** Representative images SNpc of male mice at 28 weeks post-injection of AAV-A53T-SNCA. Inset is enlarged on the right. Scale bar, 1 mm for left panels, 100 μm for right panels.

**d** Quantifications of average phospho-S129 α-synuclein fluorescence intensity in DA neurons of male mice in each group ($n = 3$ mice for empty-vector control groups; $n = 7$ mice for AAV-SNCA groups). Significance determined by ANOVA. Error bars represent mean ± s.d. ****$P < 0.0001$; ns, not significant **e** Percentage of contralateral forelimb use for rearing in the cylinder test of female (open dots) or male (closed dots) mice at 28 weeks post unilateral injection of AAV-Null or AAV-A53T-SNCA vectors ($n = 29$ for WT + EV, $n = 21$ for *mito*-QC + EV, $n = 32$ for *mito*-QC/*Usp30* KO + EV, $n = 30$ for WT + SNCA, $n = 28$ for *mito*-QC + SNCA, $n = 29$ for *mito*-QC/ *Usp30* KO + SNCA). Significance determined by one-way ANOVA. Error bars represent mean ± s.d.; ****$P < 0.0001$; ns, not significant. Source data are provided as a Source Data file.

in *mito*-QC/*Usp30* KO AAV-A53T-SNCA injected mice versus 67.10 ± 4.899 in WT or 76.62 ± 4.854 in *mito-QC* injected with AAV-A53T-SNCA, $p < 0.0001$).

To further understand whether USP30 depletion affects the association between pathological αSyn and mitochondria in the PD model, we analyzed the overlap of a mitochondrial marker (OPA-1, red) with phospho-S129-αSyn (green) in the ipsilateral SNpc of the AAV-A53T-SNCA mouse model (Supplementary Fig. 5b). The mitochondria visualized by OPA-1 staining were mostly visible as puncta in the ipsilateral SNpc of *mito*-QC/*Usp30* WT mice but were visible as a dynamic network in the *Usp30* KO mice (Suppl. Fig. 5b). The colocalization analysis showed roughly 80% of total mitochondria was associated with pathological S129-αSyn in the SNpc neurons of the AAV-A53T-SNCA -injected WT mice (78.09 ± 1.218 %) (Supplementary Fig. 5c). Intriguingly, the fraction of mitochondria that co-stained with S129-αSyn was dramatically decreased to around 20% in the *Usp30* KO mice

(16.93 ± 2.739 %) (Supplementary Fig. 5c). Furthermore, the fraction of pathological S129-αSyn located on mitochondria remains comparable in both the WT (46.39 ± 2.638 %) and *Usp30* KO mice (44.79 ± 2.617 %) (Supplementary Fig. 5d). These results show that USP30 depletion reduced the accumulation of pathological S129-αSyn on mitochondria.

## *Usp30* KO protects against αSyn-induced motor deficits

To assess if loss of dopaminergic neurons at 28 weeks post-AAV-A53T-SNCA injection is associated with motor deficits, and to assess the impact of *Usp30* KO on motor function, we measured the asymmetrical usage of forelimbs in the cylinder test, which is sensitive to asymmetric dopamine deficiency[47–51]. Unilateral injection of AAV-Ev did not affect motor function in female or male mice of all three experimental groups (WT, *mito*-QC and *mito*-QC/*Usp30* KO; Fig. 2f), thus excluding any nonspecific effect from the virus or the stereotaxic injection itself. In contrast, unilateral injection of the AAV-A53T-SNCA

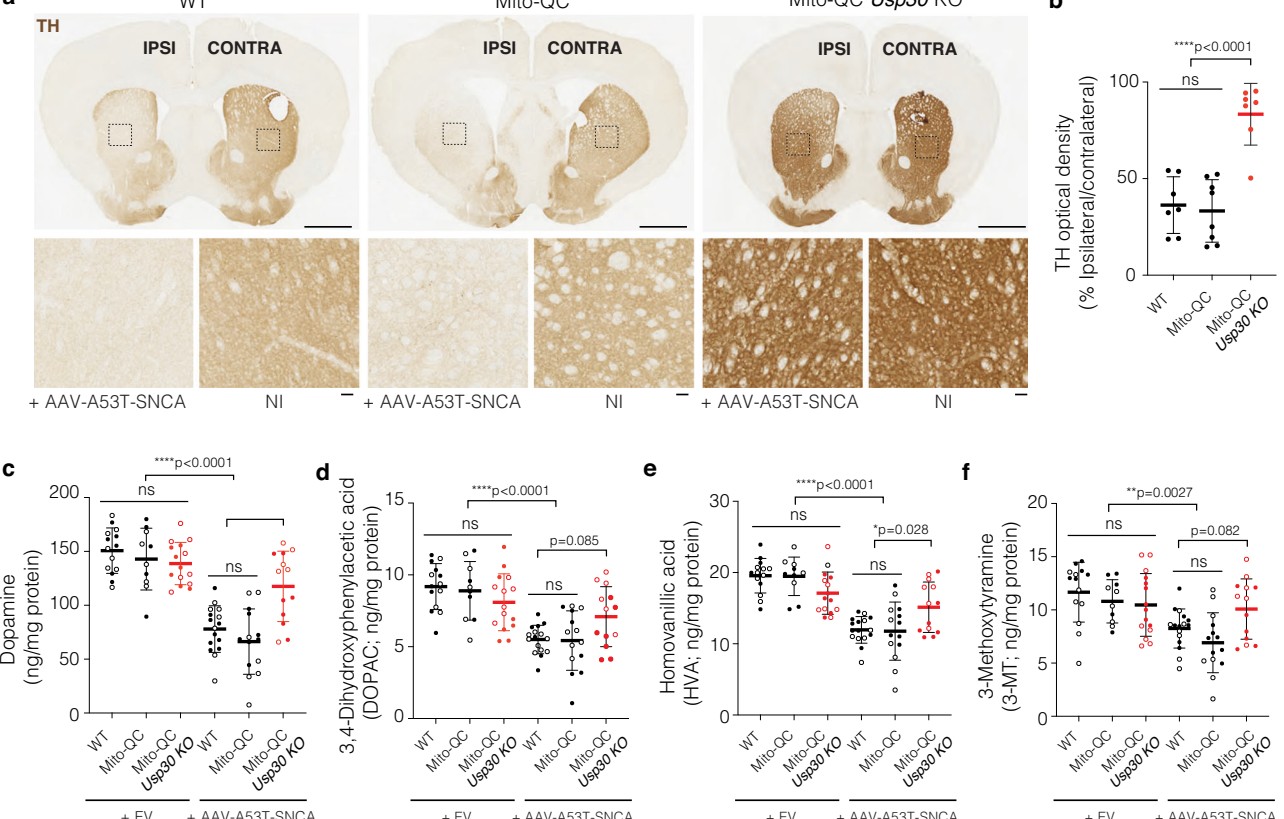

**Fig. 3 | *Usp30* KO prevents loss of striatal TH+ dopaminergic fibers and preserves dopamine and its metabolites in the α-synuclein-based mouse model.**
**a** Representative striatal sections of TH immunohistochemistry in male mice at 28 weeks post-injection of AAV-A53T-SNCA.; enlarged in lower panels. Scale bar, 1 mm for upper panels, 100 μm for lower panels. **b** Relative optical density of TH+ fibers in the ipsilateral striatum compared with the contralateral side of male mice in each group (*n* = 7 for WT group, *n* = 8 for *mito*-QC group, *n* = 7 for *mito*-QC/*Usp30* KO group). Significance determined by one-way ANOVA. Error bars represent mean ± s.d.; ****$P < 0.0001$. **c** Mean dopamine levels (ng/mg protein) in the ipsilateral striatum of female (open dots) and male (closed dots) mice in each group (*n* = 14 for WT + EV, *n* = 10 for *mito*-QC + EV, *n* = 15 for *mito*-QC/*Usp30* KO + EV, *n* = 16 for WT + SNCA, *n* = 14 for *mito*-QC + SNCA, *n* = 13 for *mito*-QC/*Usp30* KO + SNCA groups of mice). Significance determined by one-way ANOVA. Error bars represent mean ± s.d.; ***$P < 0.001$, ****$P < 0.0001$. **d** Mean of 3,4-Dihydroxyphenylacetic acid levels (DOPAC, ng/mg protein) in the ipsilateral striatum of female (open dots) and male (closed dots) mice in each group (*n* = 14 for WT + EV,

*n* = 10 for *mito*-QC + EV, *n* = 15 for *mito*-QC/*Usp30* KO + EV, *n* = 16 for WT + SNCA, *n* = 14 for *mito*-QC + SNCA, *n* = 13 for *mito*-QC/*Usp30* KO + SNCA groups of mice). Significance determined by one-way ANOVA. Error bars represent mean ± s.d.; ****$P < 0.0001$. **e** Mean levels of homovanillic acid (HVA, ng/mg protein) in the ipsilateral striatum of female (open dots) and male (closed dots) mice in each group (*n* = 14 for WT + EV, *n* = 10 for *mito*-QC + EV, *n* = 15 for *mito*-QC/*Usp30* KO + EV, *n* = 16 for WT + SNCA, *n* = 14 for *mito*-QC + SNCA, *n* = 13 for *mito*-QC/*Usp30* KO + SNCA groups of mice). Significance determined by one-way ANOVA. Error bars represent mean ± s.d.; ****$P < 0.0001$; *$P < 0.05$. **f** Mean levels of 3-methoxytyramine (3-MT, ng/mg protein) in the ipsilateral striata of female (open dots) and male (closed dots) mice in each group (*n* = 14 for WT + EV, *n* = 10 for *mito*-QC + EV, *n* = 15 for *mito*-QC/*Usp30* KO + EV, *n* = 16 for WT + SNCA, *n* = 14 for *mito*-QC + SNCA, *n* = 13 for *mito*-QC/*Usp30* KO + SNCA groups of mice). Significance determined by one-way ANOVA. Error bars represent mean ± s.d.; **$P < 0.01$. Source data are provided as a Source Data file.

in both WT and *mito*-QC mice induced motor dysfunction, as shown by less use of the forelimbs contralateral to the injection (compared to use of the ipsilateral forelimb) (Fig. 2e). Notably, *Usp30* KO significantly protected against the αSyn-induced motor deficits in both female and male *mito*-QC/*Usp30* KO mice (*mito-QC* AAV-A53T-SNCA versus *mito*-QC/*Usp30* KO AAV-A53T-SNCA, *p* < 0.0001; WT AAV-A53T-SNCA versus *mito*-QC/*Usp30* KO AAV-A53T-SNCA, *p* < 0.0001; Fig. 2e and Supplementary Videos 1–3). These results demonstrate that *Usp30* KO rescues αSyn-induced motor deficits, as measured by the cylinder test.

### *Usp30* KO protects against αSyn-induced loss of striatal dopamine and TH+ terminals

To test the impact of *Usp30* KO on αSyn-induced loss of DA neurites projecting into the striatum, we measured the density of TH+ terminals in the striatum of brain sections (Fig. 3a). The relative optical density of TH+ fibers was significantly decreased in both WT mice (36.28 ± 5.539 %; *p* < 0.0001, Fig. 3b) and mito-QC mice (33.26 ± 5.721

%; *p* < 0.0001, Fig. 3b), but not in the *Usp30* KO mice (84.05 ± 5.277 %; Fig. 3b) following AAV-A53T-SNCA injection.

Upon TH staining we also observed that TH-intensity both in striatum and SNpc differed from WT to *Usp30* KO even in the non-injected side. To understand why, we quantified TH optical density in mice that were either injected with AAV-A53T-SNCA or injected with AAV-Empty vector (Ev) in the ipsilateral side in relation to the NI contralateral side. These data showed that in the AAV-A53T-SNCA injected mice, TH staining intensity was affected not only on the ipsilateral side but also on the contralateral side (Fig. 3a, b and Supplementary Fig. 4c, d). The bilateral effect on striatal TH+ terminals following unilateral injection of an AAV-vector to overexpress αSyn in SN has been reported in a rat PD model[53]. Thus, TH staining likely is not elevated directly by *Usp30* KO mice, and instead *Usp30* KO protects against the aSyn-induced loss of TH that occurs on both the ipsilateral and contralateral sides. This observation does not alter our overall data interpretation as, for our analyses, we compare intra-mouse effects (Ipsiateral Vs. Contralateral).

We further analyzed the molecular levels of dopamine and its metabolites in the ipsilateral striata of both female and male mice. The levels of dopamine, and its metabolites, including 3,4-dihydroxyphenylacetic acid (DOPAC), 3-methoxytyramine (3-MT) and homovanillic acid (HVA), are comparable across genotypes in AAV-Ev injected mice (Fig. 3 c-f). AAV-A53T-SNCA injection caused dopamine depletion in both WT and *mito*-QC mice ($p < 0.0001$; Fig. 3c) but not in *Usp30* KO mice (Fig. 3c). Furthermore, *Usp30* KO prevented the decline of dopamine metabolites HVA ($p < 0.05$, Fig. 3e) and 3-MT ($p = 0.0082$, Fig. 3f), with a nonsignificant trend for DOPAC ($p = 0.085$, Fig. 3d). These results show that *Usp30* KO protects against loss of TH+ striatal terminals and against striatal dopamine loss in this αSyn-based mouse model of PD.

## Validation of a brain penetrant USP30 Inhibitor MTX115325

MTX115325 is a proprietary USP30-inhibitor (USP30i) developed by Mission Therapeutics with good oral bioavailability and central nervous system (CNS) penetration (Fig. 4, Supplementary Fig. 6a) (WO 2021/249909 A1). MTX115325 (Fig. 4a) inhibits USP30 in a biochemical fluorescence polarization assay with an IC50 of 12 nM (Suppl. Fig. 6a, first row) and, in cells, blocks access of a ubiquitin-like probe to the enzyme active site with an IC50 of 25 nM (Supplementary Fig. 6a, second row). In a human HeLa cell line overexpressing PARKIN and challenged with the mitochondrial toxins antimycin A and oligomycin A (A/O) MTX115325 increased ubiquitylation of the outer mitochondrial membrane protein TOM20, a USP30 substrate, with an EC1.5x and EC50 of 10 nM and 32 nM respectively (Supplementary Fig. 6a, third row; a representative TOM20-Ub concentration-response profile is provided in Fig. 4b). To investigate further pharmacological effects of MTX115325 in human dopaminergic neurons in vitro, iPSC-derived dopaminergic neurons (control or alpha-synuclein A53T mutated iPSC-derived dopaminergic neurons) were tested for differences in TOM20 and TOM20-ubiquitylation, in the absence or presence of 7 days MTX115325 treatment at 10 nM, 100 nM or 1 μM without any exogenous stimuli. MTX115325 treatment caused the upregulation of TOM20-ubiquitylation (Supplementary Fig. 6b, d–b contains western blot data from control neurons). Shorter term 24 h treatment with 1 μM MTX115325 without any exogenous stimuli also increased TOM20-ubiquitylation in both control and A53T αSyn backgrounds (Fig. S6c).

To assess selectivity against DUBs and other cysteinyl proteases, the USP30i MTX115325 was screened against 54 DUBs and five cathepsins in biochemical assays (Table 1). The highest level of DUB inhibition was IC50 24.9 μM, and the highest cathepsin was for Cathepsin L at IC50 42.1 μM, representing a > 2000 fold and >3500-fold USP30 selectivity respectively (Supplementary Fig. 6a, fourth row and Table 1). Importantly, for use in mouse in vivo studies, we confirmed that the USP30i MTX115325 inhibited mouse USP30 with an IC50 of 13 nM, demonstrating equivalent potency between the human and mouse enzymes (Supplementary Fig. 6a, fifth row). Pharmacokinetic studies in mice were performed via intravenous (IV) delivery to measure compound clearance, via oral delivery to measure bioavailability, and via oral delivery in animals with a micro-dialysis probe inserted into the prefrontal cortex to measure free drug profile in the brain interstitial fluid. These analyses revealed that MTX115325 exhibits excellent oral bioavailability of 98% (Supplementary Fig. 6a, sixth row), low to moderate metabolic clearance of 19.7 mL/min/kg (Supplementary Fig. 6a, seventh row) and good CNS penetration with an unbound partitioning coefficient, Kpu,u of approximately 0.4 (Supplementary Fig. 6a, eigth row,). A time concentration profile is provided of the USP30i MTX115325 in whole blood and prefrontal cortex dialysate following a single 10 mg/kg dose, demonstrating a tight relationship of both tissue profiles (Fig. 4c), and a prefrontal cortex free CMax of 528 nM (at the 60-min time point). In a separate study, MTX115325 demonstrated good CNS target engagement with 10 mg/kg PO dosing achieving approximately 8 h of 50% CNS USP30 binding

(Supplementary Fig. 6e). Mouse exploratory toxicology studies demonstrated that MTX115325 is well tolerated with no adverse clinical observations or pathology findings after two weeks of dosing with dose levels up to 300 mg/kg/day. In SH-SY5Y cells that stably express *mito*-QC, MTX115325 produced a concentration-dependent increase in mitophagy when incubated for 72 h in combination with a submaximal inhibition of ETC Complex III and V (0.1 mM A/O) (Fig. 4e). MTX115325 increased mitophagy by 22% at 37 nM with a maximum increase of 54% compared to baseline at 1 μM (Fig. 4f). These data show that inhibition of USP30 with MTX115325 is highly selective, CNS penetrant and well tolerated and that MTX115325 drives mitochondrial quality control processes in a neuroblastoma cell line and in IPSC-derived neurons in vitro with concentration-dependent effects in the presence and absence of exogenous stimuli.

## The USP30 inhibitor MTX115325 prevents dopaminergic neuronal loss and preserves striatal dopamine in an AAV-A53T-SNCA mouse model

To test whether inhibition of USP30 catalytic activity can recapitulate the protective effects on the nigrostriatal system observed in *Usp30* KO mice, we tested the USP30i MTX115325 (WO 2021/249909 A1) in a closely related version of the AAV-A53T-SNCA PD mouse model as we described for the *Usp30* KO studies. Following unilateral stereotaxic injection of an AAV1/2 encoding AAV-A53T-SNCA and no contralateral injection (NI), mice were treated twice daily with 15 mg/kg or 50 mg/kg of MTX115325 by oral gavage, and after 10 weeks of treatment ipsilateral and contralateral striatum were harvested for measurement of dopamine and metabolites, and SN harvested for measurement of TH+ neurons (50 mg/kg group only). Example TH staining images are provided in Fig. 5a. Consistent with the effect of *Usp30* KO described above, USP30 inhibition with MTX115325 protected against A53T αSyn induced loss of TH+ neurons (Fig. 5b). Using each animal's contralateral NI hemisphere as their own control, the percentage of ipsilateral *vs* contralateral TH+ neurons in vehicle-treated animals was 61.7% vs 89.08% for MTX115325 (50 mg/kg BID) treated animals with a p-value of 0.029 (unpaired Student's t-test), indicating significant protection against αSyn induced neuronal loss from pharmacological inhibition of USP30. Consistent with these effects on TH+ neurons, MTX115325, at both 50 and 15 mg/kg BID, abrogated loss of dopamine and dopamine metabolites HVA and DOPAC in the ipsilateral hemisphere compared to the contralateral hemisphere, as well when comparing levels in the ipsilateral hemisphere between treatment groups (Fig. 5c–e). When comparing levels in the ipsilateral hemisphere between treatment groups using normalized data, although mean dopamine/HVA/DOPC estimates suggest slightly less robust efficacy at 15 mg/kg BID compared to 50 mg/kg BID, they are insufficient to definitively establish a dose responsive effect of MTX115325 treatment in this model.

Additionally, mechanistic markers of disease-relevant pathology were assessed including total αSyn, phosphorylated S129-αSyn and microglial/astrocyte abundance (Iba-1/GFAP). USP30 inhibition with MTX115325 at 50 mg/kg significantly decreased total GFAP stained area (Supplementary Fig 7b) reflecting lower numbers of activated astrocytes. MTX115325 at 50 mg/kg significantly reduced phosphorylated S129-αSyn but not total αSyn (Supplementary Fig. 7c, d).

To confirm compound exposure consistent with robust levels of target engagement in the study, MTX115325 levels were measured in blood samples taken at 0.5, 1, 2, 4 and 6 h after the first dose, seven days before the end of the study. MTX115325 achieved a blood Cmax of 7546.9 ng/mL at 15 mg/kg and 16374.3 ng/mL at 50 mg/kg and exposures of 13606 ng*h/mL and 42959 ng*hr/mL, respectively, at the different doses (Fig. 4d). Estimated brain free drug concentrations would be well above the EC50 in the TOM20 ubiquitylation (TOM20-Ub) assay for the duration of the dosing regimen at the 50 mg/kg dose level.

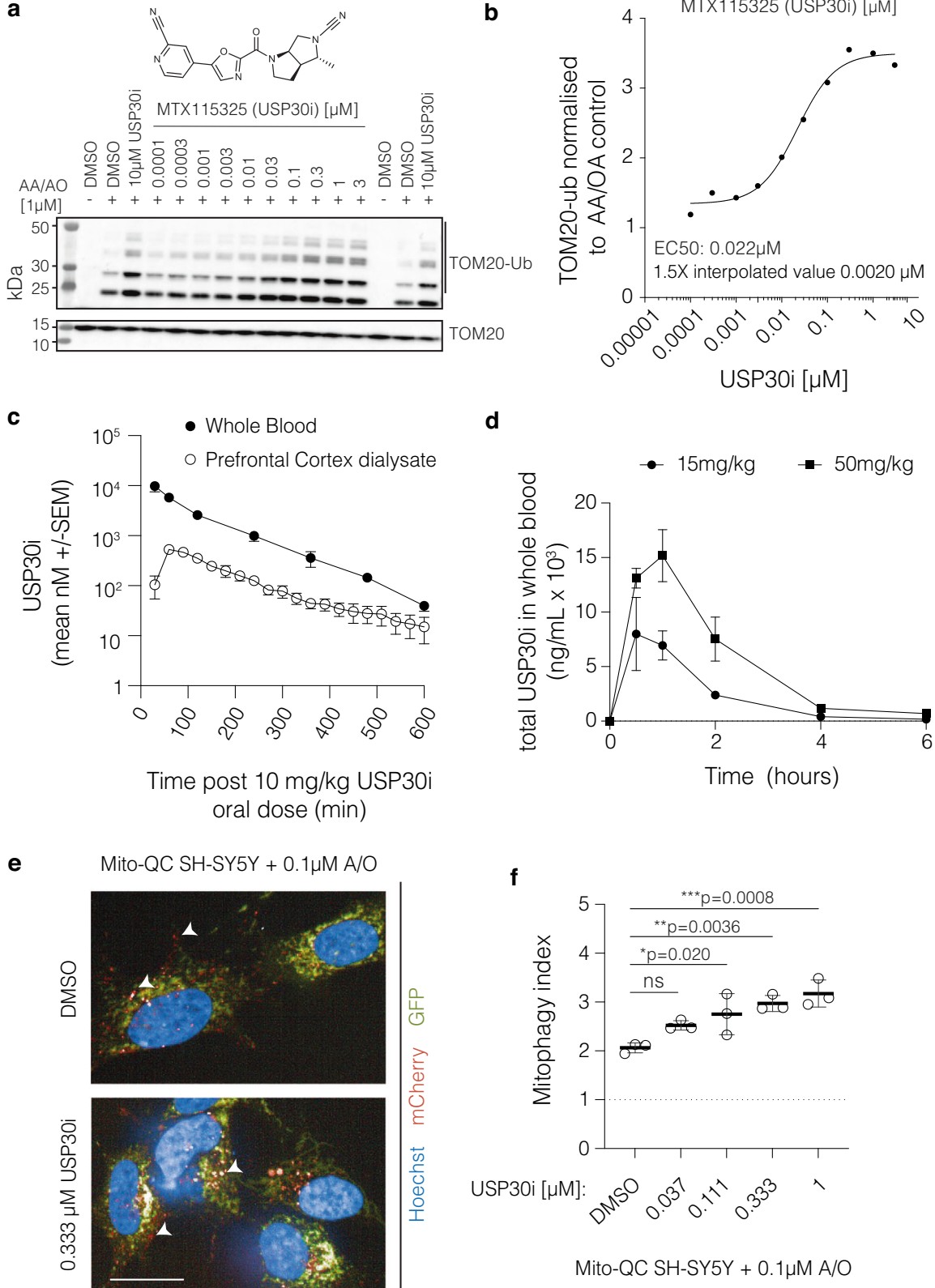

**Fig. 4 | Validation of USP30 inhibition and pharmacokinetics of a small molecule USP30 inhibitor, MTX115325. a** Representative Western blot images from 5 independent experiments show TOM20 and Ubiquitin-modified TOM20 (TOM20-Ub) at various concentrations of MTX115325 for 90 min in HeLa cells. The structure of MTX115325 is shown above. **b** Quantification of normalized TOM20-Ub at various MTX115325 concentrations. **c** Time-dependent concentration of MTX115325 in whole blood and prefrontal cortex after oral administration at 10 mg/kg. *n* = 4 mice per group. **d** Whole blood concentrations of MTX115325 after oral administration of 15 mg/

kg and 50 mg/kg in the A53T model. *n* = 3 mice per group. **e** Representative images showing *mito*-QC signals in SHSY-5Y cells, which were counterstained with Hoechst for nuclei (blue), after exposing to MTX115325 at 0.333 μM for 20 min. Scale bar, 20 μm. The uncropped images are presented in Supplementary Fig. 9. **f** quantification of the mitophagy index with MTX115325 treatment (*n* = 3 independent experiments, with three technical replicates capturing 11 fields of views). Statistical analysis using one-way ANOVA with post-hoc Dunnett's test. Error bars represent mean ± s.d.;*P < 0.05; **P < 0.01; ***P < 0.001. Source data are provided as a Source Data file.

**Table 1 | Selectivity of MTX115325 USP30i**

| Enzyme | IC50 (μM) |
|---|---|
| USP2 | 24.9 |
| JOSD1 | 31.0 |
| Cathepsin L | 42.1 |
| USP21 CD | 60.7 |
| USP16 | 83.7 |
| USP25 | 93.2 |
| USP6 | 96.3 |
| USP28 | 106.1 |
| USP10 | 107.2 |
| Cathepsin K | 109.9 |
| USP32 | 120.2 |
| USP22 | 147.3 |
| USP36 CD | 157.3 |
| JOSD2 | 159.5 |
| ATXN3L | 172.5 |
| Cathepsin B | 205.1 |
| USP47 | 212.5 |
| Cathepsin V | 247.5 |
| Cathepsin S | 259.6 |
| USP19 | 277.5 |
| USP11 | 290.9 |
| USP8 | >300 |
| USPL1 | >300 |
| MINDY3 | >300 |
| USP24 | >300 |
| VCPIP FL | >300 |
| OTUD6A | >300 |
| USP48 | >300 |
| OTUB1/UBCH5B | >300 |
| YOD1 | >300 |
| CYLD | >300 |
| CEZANNE1 | >300 |
| SENP2 CD | >300 |
| UchL5 | >300 |
| USP4 | >300 |
| USP12/UAF1 | >300 |
| USP34 CORE | >300 |
| OTUD1 CD | >300 |
| USP35 | >300 |
| UchL1 | >300 |
| UchL3 | >300 |
| USP9x | >300 |
| USP46/UAF1 | >300 |
| SENP1 CD | >300 |
| OTUB2 | >300 |
| USP7 | >300 |
| BAP1 | >300 |
| OTUD5(p177S) | >300 |
| USP13 | >300 |
| MINDY2 | >300 |
| OTUD3 CD | >300 |
| USP15 | >300 |
| USP14 | >300 |
| USP1 | >300 |
| Trabid | >300 |

**Table 1 (continued) | Selectivity of MTX115325 USP30i**

| Enzyme | IC50 (μM) |
|---|---|
| USP20 | >300 |
| USP5 | >300 |
| OTUD6B | >300 |
| SENP6 | >300 |

Taken together, these results show that USP30 inhibition by MTX115325 recapitulates the effects of *Usp30* KO in protecting against TH+ neuronal loss and striatal dopamine loss in an αSyn-based PD mouse model.

## Discussion

The current study investigated the effects of *Usp30* loss or inhibition with a potent and USP30 selective inhibitor, MTX115325, in an αSyn-driven chronic dopaminergic degeneration model. We show that in vivo USP30 inhibition, through genetic removal or small molecule inhibition, represents a viable strategy for enhancing the clearance of damaged mitochondria through mitophagy in this αSyn-based mouse PD model.

We targeted USP30 to experimentally control mitophagy levels through the genetic knockout of *Usp30* and by pharmacological inhibition of USP30 catalytic activity. USP30 is a deubiquitylase, which counteracts the effects of PARKIN by removing ubiquitin from mitochondrial outer membrane proteins[39,54]. USP30 cysteinyl protease catalytic activity prefers Lys6-linked Ubiquitin chains[43]. Recent studies have extensively mapped outer mitochondrial membrane (OMM) substrates, which include several members of the TOM (translocase of outer mitochondrial membrane) family proteins, VDAC (voltage-dependent anion-selective channel) family proteins, CISD1 (CDGSH iron sulfur domain 1), and FKBP8 (FKBP prolyl Isomerase 8) amongst others[41,42]. Knockout of the *Usp30* gene upregulates mitophagy and increases the clearance of damaged mitochondria in induced-neurons derived from embryonic stem cells and in SH-SY5Y neuroblastoma cells[41,42]. Of interest, given the strong link between TOM20 and mitochondrial-derived vesicle pathways[55,56], USP30 may interact with mitochondrial-derived vesicles (MDVs) to elicit further mitochondrial quality control[57]. USP30 may also play a role in regulation of peroxisome abundance/turnover[58]. USP30 may affect mitochondrial function through mitophagy and the translation of mitochondrial proteins in cytotoxic T lymphocytes (CTL)[59]. We have not explored either of these mechanisms in the current study and cannot exclude their potential role in the beneficial outcomes for *Usp30* KO/inhibition. Other DUBs, including USP13, USP14, USP15, USP33, and USP35, also have been suggested to have antagonizing effects on PARKIN-mediated ubiquitylation and mitophagy and additional E3 ligases are known to be able to ubiquitylate mitochondria[54,60–64], although with limited consensus across multiple independent groups.

We found that knockout of *Usp30 gene* in mice led to increased basal levels of mitophagy in dopaminergic neurons in the SN. Our results on the *Usp30* KO mice align with data from Phu et al.[1], which showed that USP30 depletion accelerated mitophagy and led to increased basal respiration but reduced reserve capacity in hippocampal neurons, highlighting that USP30 plays a crucial role in regulating mitochondrial homeostasis and its absence influences mitochondrial metabolism and mitophagy in neurons.

Overexpression of αSyn induces synucleinopathy in dopaminergic neurons and affects mitochondrial function and thus is a suitable model to test if a mitophagy regulation strategy could protect against αSyn toxicity. Mitochondrial dysfunction plays a central role in the pathogenesis of dopaminergic neurodegeneration in PD. The products

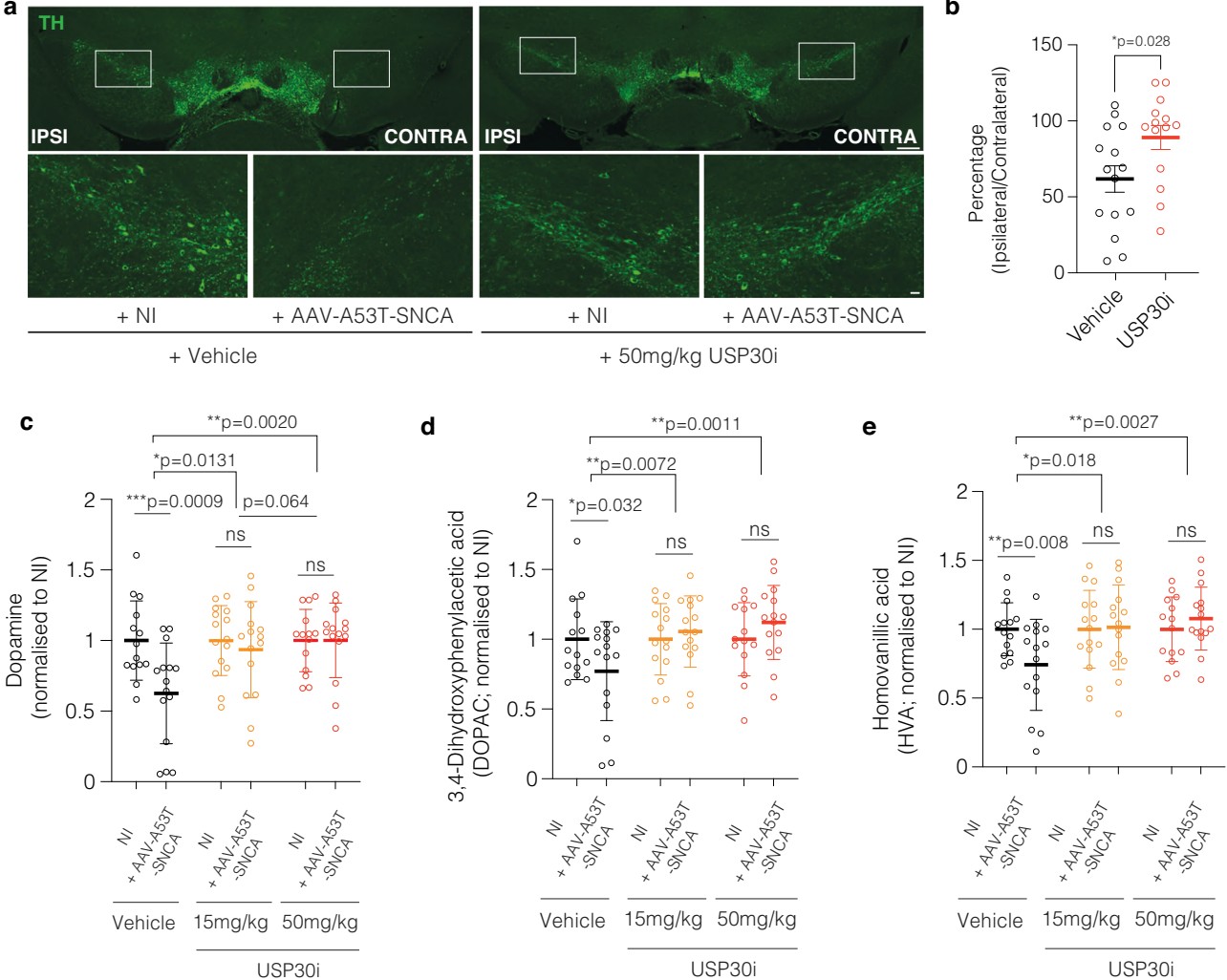

**Fig. 5 | Pharmacological inhibition of USP30 with MTX115325 prevents dopaminergic neuronal loss and dopamine depletion in an α-synuclein-based PD mouse model. a** Representative immunofluorescence images of TH in the SNpc of male mice. Inlets enlarged in lower panels. Scale bar, 1 mm for upper panels, 10 μm for lower panels. **b** Percentage of dopaminergic neurons in the AAV-A53T-SNCA injected side versus non-injected (NI) side. $n = 15$ for vehicle group, $n = 14$ for USP30i group. Significance determined by unpaired $T$ test (2-tailed), Error bars represent mean ± s.d. *$P < 0.05$. **c** Relative dopamine levels in male mice's striatum at 10 weeks post-injection ($n = 14, 15, 15, 15, 14, 15$ for vehicle+NI, vehicle+SNCA, USP30i 15 mg/kg+NI, USP30i 15 mg/kg+SNCA, USP30 30 mg/kg+NI, USP30 30 mg/kg+SNCA groups respectively). Error bars represent mean ± s.d. *$P < 0.05$;

**$P < 0.01$; ***$P < 0.001$. **d** Relative DOPAC levels in male mice's striatum at 10 weeks post-injection ($n = 14, 15, 15, 15, 14, 15$ for vehicle+NI, vehicle+SNCA, USP30i 15 mg/kg+NI, USP30i 15 mg/kg+SNCA, USP30i 30 mg/kg+NI, USP30i 30 mg/kg+SNCA groups respectively). Error bars represent mean ± s.d. *$P < 0.05$; **$P < 0.01$. **e** Relative levels of HVA in the male mice's striatum at 10 weeks post-injection ($n = 14, 15, 15, 15, 14, 15$ for vehicle+NI, vehicle+SNCA, USP30i 15 mg/kg+NI, USP30i 15 mg/kg+SNCA, USP30i 30 mg/kg+NI, USP30i 30 mg/kg+SNCA groups respectively). Significance for ipsi vs contra comparisons determined by 2-way ANOVA with Uncorrected Fisher LSD. Significance for ipsi vs ipsi comparisons determined by 2-way ANOVA with Dunnett's post-hoc test. Error bars represent mean ± s.d.; *$P < 0.05$; ***$P < 0.01$. Source data are provided as a Source Data file.

of several familial PD-related genes, including PARKIN, PINK1, LRRK2, DJ-1, and UCHL-1, directly regulate mitochondrial homeostasis[44,65–68]. Parkinsonism-related neurotoxins, including 1-methyl-4-phenyl-1,2,3,6-tetrahydropyridine (MPTP), rotenone, paraquat and 6-hydroxydopamine, directly inhibit mitochondrial function and ultimately decrease neuronal viability[69]. αSyn binds to the mitochondrial membrane through its lipophilic N-terminus, impairs mitochondrial permeability transition pores (mPTP) and decreases mitochondrial membrane potential and cellular viability[70]. αSyn also binds to TOM20 on the mitochondrial membrane and blocks protein import[24]. Accumulation of WT and A53T αSyn and decreased mitochondrial complex I activity are found in human fetal dopaminergic neurons that overexpress αSyn[71]. Moreover, accumulated αSyn is found on mitochondria isolated from SN and striatum of PD patients but not in other brain regions or brain samples from healthy controls[71], and is also detected in the SN, putamen, amygdala and hippocampus on brain sections from

Lewy body disease (LBD) patients[72]. In addition, small αSyn oligomers induce fission protein DRP1-independent mitochondrial fragmentation in neurons in vitro and in vivo[73]. Overexpressed human A53T αSyn results in extensive mitochondrial damage and loss, which leads to bioenergetic deficits and neurodegeneration in primary neuronal culture[74]. More recently, iPSC-derived neurons harboring A53T mutations from PD patients were used to demonstrate oligomerization and subsequent αSyn aggregation triggered by mitochondrial lipids (cardiolipin) and by mitochondrially generated ROS[75]. These findings indicate that αSyn induces mitochondrial dysfunction, leading us to test a strategy for enhancing mitophagy through USP30 removal/inhibition as a potential neuroprotective strategy in a slowly degenerative αSyn-based mouse model of PD.

We have further tested these mice for vulnerability to αSyn toxicity by using an AAV-A53T-SNCA overexpression mouse model that recapitulates key PD features, including motor deficits, dopamine

depletion in the striatum and synucleinopathy, chronically at 28 weeks post-injection of the vector. *Usp30* KO successfully prevented the development of αSyn-induced motor deficits in both female and male mice with AAV-A53T-SNCA injections. The protection of motor function in *Usp30* KO mice was consistent with attenuated loss of TH+ neurons in the SNpc, preservation of dopamine and its metabolites in the striatum, significantly decreased accumulation of phospho-S129 αSyn and enhanced clearance of phospho-S129 αSyn impaired mitochondria in dopaminergic SNpc neurons. These data collectively demonstrate that absence of USP30 in the *Usp30* KO mice leads to enhanced mitophagy and potent protection against αSyn toxicity.

Regarding other USP30 inhibitors[42,76], a N-cyano pyrrolidine compound, FT3967385, was found to significantly increases the ubiquitylation of TOM20 and mitophagy levels, which is comparable to the effects of genetic loss of the *Usp30* gene in SHSY5Y neuroblastoma cells, supporting the notion that the catalytic activity of USP30 is important for modulating mitophagy[42]. While some synthetic racemic phenylalanine derivatives, MF-094 and MF-095[77], and a benzosulphonamide inhibitor, CMPD-39[78], also have been identified as USP30 inhibitors in biochemical assays[77,78], no in vivo activity of any of these molecules has been reported. This study describes the potent and selective USP30 inhibitor, MTX115325, a highly selective USP30 inhibitor with good drug-like properties, in vivo CNS penetration and a good exploratory toxicology profile. As expected, MTX115325 activates mitophagy in SH-SY5Y cells in vitro, to a similar magnitude as observed by other USP30 inhibitors[42]. We have used MTX115325 to test the effect of the inhibition of USP30 catalytic activity to complement data generated in *Usp30* KO mice. Inhibition of the catalytic activity of USP30 with MTX115325 leads to robust protection of TH+ neurons in the SNpc and preservation of dopamine and its metabolites in the striatum, consistent with the protective effects resulting from KO of *Usp30*. Our results highlight that inhibition of the USP30 catalytic activity is promising as a potential therapeutic strategy for PD. Notably, extensive phenotyping of *Usp30* KO mice (see Supplementary Data 1) revealed no adverse overt pathologies in the *Usp30* KO mice, and in contrast, there were potential health benefits against age-related fat accumulation in liver, which is consistent with another study shows that USP30 depletion attenuates lipogenesis and protects against tumorigenesis in the liver[79].

Despite these interesting results, the study does have some limitations and future directions that we would like to investigate. We did not deploy the stereological investigator system for TH+ neuronal counting. The QuPath software can identify all positive neurons in a designated brain region and counts all neurons efficiently and in an unbiased manner[80–82]. Prior data demonstrates that the stereological estimation of dopaminergic neurons in the SNpc of mouse brain with ImageJ software using the method that we employed is comparable with conventional stereological counting with the optical fractionator method[83]. In the genetic model, we note that expression of mutant αSyn did not affect the basal level of mitophagy independent of USP30 loss; this may also be partially due to changes happening before our selected point of analysis and that we cannot account for in a chronic model that spans 28 weeks for mitophagy analysis. In the USP30i MTX115325 studies, only TH+ cells with nuclei were counted. Increased mitophagy demonstrated in *Usp30* KO mice and with MTX115325 in vitro raises the possibility that mitophagy may be partly responsible for the impact of USP30 inhibition on reducing αSyn pathology in vivo, potentially through the preferential degradation of mitochondria with S129-αSyn bound to the surface. However, we have not established the mechanisms of this effect; for example, there is evidence that mitochondrial quality may impact αSyn aggregation properties through cardiolipin concentration in lipid membrane[75] and PINK1 activator in αSyn preformed fibrils (PFFs) models[84]. Furthermore, USP30 may in part be localized to peroxisomes and may play a role in pexophagy[58,78,85], although a role of pexophagy in PD has not been

established. USP30 affects the homeostasis of mitochondria in the cytotoxic T lymphocytes (CTL) and is important for sustaining killing capacity of CTL against target cells[59]. Finally, potential interactions of *Usp30* KO with other PD-related genes are unknown and should be explored in the future.

In conclusion, we used a strategy for upregulating mitophagy by *Usp30* deletion, with similar results obtained with a pharmacological USP30 inhibitor. These strategies to reduce USP30 lead to enhanced mitophagy and potent protection against αSyn toxicity. This work validates inhibition of USP30 as a promising strategy for further testing for potential disease-modifying effects in PD.

## Methods
### Study design
The primary goal of this study was to investigate the effects of *Usp30* knockout or pharmacological inhibition in an αSyn based PD mouse model. Previous studies have shown the role of USP30 in regulating mitophagy in cells[39–41,58] and the protective effects of *Usp30* knockdown in a PARKIN-deficient drosophila model[39]. The current study demonstrates the role of USP30 in regulating the mitophagy pathway in mouse brain. Further, it shows the protective effects of *Usp30* knockout against α-synucleinopathy and dopaminergic neurodegeneration in an αSyn based mammalian model. The AAV1/2-A53T aSyn vector was used to overexpress human A53T mutant αSyn to induce progressive synucleinopathy and dopaminergic neurodegeneration in the mouse brain. The sample size of the study and the endpoints of the experiments were based on previously published studies and our preliminary results. Age-matched littermates were randomly assigned to treatment or control groups. Both females and males were included in the study, and statistical analysis was conducted by following NIH's 2015 sex as a biological variable (SABV) policy[86–88]. Drug dosage was determined as previously described[71]. All analyses were done blindly.

### Animals
**Ethical information.** Mouse studies at the Wellcome Sanger Institute (WSI) were performed in accordance with UK Home Office regulations and the UK Animals (Scientific Procedures) Act of 2013 under UK Home Office licenses. The WTSI Animal Welfare and Ethical Review Board approved these licenses. Mouse studies at BIDMC were approved by the local Institutional Animal Care and Use Committee (IACUC, animal protocol number 018-2019). At Charles River Finland laboratories (animal protocol number C0610721), the mouse work was done in accordance with all applicable national, international, and/or institutional guidelines for the care and use of animals (Finnish national legislation): (1) Act on the Protection of Used for Scientific or Educational Purposes (497/2013) (2) Government Decree on the Protection of Animals Used for Scientific or Educational Purposes (564/2013) (3) License for animal experiment approved by National Animal Experiment Board: 18537-2018. European and international legislation and guidelines: (1)Directive 2010/63/EU (2) Commission recommendation 2007/526/EC (3) Guide for the Care and Use of Laboratory Animals (Guide), Eighth Edition (National Research Council 2011).

**Housing and husbandry.** At BIDMC, all mice, including *mito*-QC homozygous (QC) mice and *mito*-QC/*Usp30* knockout homozygous (QC/*Usp30* KO) mice and wildtype (WT) littermates were bred and housed under a 12-h light/12-h dark cycle. The ambient temperature was 21 ± 2 °C, and the humidity 55 ± 10%. The GFP-mCherry knock-in mutation in the QC line and the *Usp30* knockout was confirmed with genotyping of ear notch tissue at 21 days old. A total of 183 mice ($n = 11$–16 for 12 groups of 12 weeks old mice, 6 male and 6 female groups: WT-Null, WT-SNCA, QC-Null, QC-SNCA, QC/KO-Null, QC/KO-SNCA) for stereotaxic injection and 27 male mice (13 *mito*-QC mice and 14 *mito*-QC/*Usp30* KO mice, 20 weeks old) for mitophagy analysis were used for this study. Mice were euthanized by carbon dioxide inhalation

at BIDMC. USP30i MTX115325 compound experiments were performed under contract at Charles River Finland laboratories. At WSI, mice were maintained in a specific pathogen-free unit on a 12-h light and 12-h dark cycle with lights off at 19:30 and no twilight period. The ambient temperature was 21 ± 2 °C, and the humidity 55 ± 10%. Mice were housed using a stocking density of 3–5 mice per cage in individually ventilated caging (Tecniplast, Sealsafe 1284 L), receiving 60 air changes per hour. In addition to Aspen bedding substrate, standard environmental enrichment of two Nestlets, a cardboard fun tunnel and three wooden chew blocks were provided. Mice were given water and diet *ad libitum*. Animals were euthanized by deeply anesthetizing with pentobarbital (180 mg/kg, i.p.).

**Mouse generation and phenotyping.** All mice are on the C57BL/6 N genetic background. *Usp30* KO mice were initially phenotyped as part of the standardized pipelines from the Mouse Genetics Project (MGP) at the Wellcome Sanger Institute (WSI)[89]. The approach (see Fig. 1) was to generate a "Knockout-first allele". This strategy relied on identifying the *Usp30* exon 4 common to all transcript variants, upstream of which a *LacZ* cassette was inserted to make a constitutive knockout/"gene-trap" known as a *tm1a* conditional allele. The constitutive *Usp30* KO allele (*tm1b*) was created by a frame-shift mutation upon Cre-mediated deletion of the *LoxP* sites flanking exon 4 by using soluble CRE protein on the mouse zygote.

The high-throughput phenotyping screen consist of standardized tests conducted according to standard operating procedures (SOPs) on all mice entering the screen. An extensive range of biological areas was assessed, including metabolism, cardiovascular, neurological, and behavioral systems, bone, sensory, and hematological systems, as well as plasma chemistry. SOPs are available at IMPReSS (www. mousephenotype.org/impress)[2]. Where possible, variables were standardized based on factors predicted to affect them. Nevertheless, measures were taken to reduce potential biases, such as the impact of different people performing the test (known as the "minimized operator") and the time of day of the trial, as defined by Mouse Experimental Design Ontology (MEDO)[90]. The data captured with the MEDO ontology can be found at http://www.mousephenotype.org/about-impc/arrive-guidelines. Data inclusion/exclusion criteria were also standardized by defining pre-established reasons for QC failures (e.g., insufficient sample, equipment error). To audit QC-failed data, all discarded data was retained and tracked in the database. Wildtype (control) mice were phenotyped at regular intervals based on their age, sex, and strain. We generated at least seven homozygote mice per sex for phenotyping. The statistical analysis took into account all variables and was conducted by a linear mixed-effects model and is presented in the Supplementary Data 1 file.

## Cell culture

SH-SY5Y cells stably expressing mCherry-GFP-Fis1101-152 reporter (SH-SY5Y *mito*-QC) kindly provided by Dr I Ganley (MRC PPU, Dundee) were cultured in DMEM/F12 media supplemented with 1% L-glutamine, 1% penicillin/streptomycin, 1% non-essential amino acids, 10% fetal bovine serum at 37 °C and 5% $CO_2$.

## Dopaminergic neuron culture

Donor-derived DopaNeuron lines were generated from source material provided by the Parkinson's Progression Markers Initiative (PPMI) through The Michael J. Fox Foundation. Human iPS cells-derived DopaNeurons were purchased from Fujifilm/Cellular Dynamics. Apparently healthy normal (AHN) (catalog # R1088) or alpha-synuclein A53T mutated (A53T) (catalog # R1109) iPSC-derived dopaminergic neurons (dopaneurons) were thawed from liquid nitrogen. 400 000 viable (counted using a cell counter) dopaneurons (counted using a cell counter) were plated per well into 24 well plates on day 0 in complete maintenance medium (CMM).

A half-medium change was performed on the cells on Day 3 with CMM. On Day 5 cells were half-medium changed with complete brainphys medium. Cells were treated on Day 7 (for cells 2 week studies) or Day 14 for 7 -day MTX115325 treatments, when a full medium change was carried out with Brainphys medium + DMSO or MTX115325 (0.01, 0.1 or 1 μM). Cells were given half medium changes with or without compounds (depending on whether treatments with compound had started) every 2-3 days.

## Materials

AAV1/2-CMV/CBA-Human A53T αSyn-WPRE-BGH-polyA (GD1001-RV) and AAV1/2-CMV/CBA-Empty-WPRE-BGH-polyA (GD1004-RV) vectors were purchased from Vigene Biosciences, pAM/SAR-CBA-Human αSyn(A53T)/HA-WPRE-BGH-polyA (for compound study) was purchased from GeneDetect Ltd. The following antibodies were used for immunostaining or immunoblotting; anti-GFP (Aves Labs Inc, catalog # GFP-1010, lot # GFP3717982, 1:500 dilution), anti-mCherry (EMDmillipore, catalog #AB356482, lot # 3249537,1:500 dilution), anti-TH (KO studies: EMDmillipore, catalog # AB152, lot # 3870479, 1:1000 dilution; compound study: Abcam, #ab76442, lot # GR3393939-2, 1:1000 dilution), anti-USP30 (Santa Cruz Biotechnology, clone B-6, catalog # sc-515235, lot # C2620, 1:1000 dilution, validation of *Usp30* KO; MRC PPU Reagents and Services, #S746D, lot # 4, 1:600 dilution, cellular ubiquitin probe binding assay), anti-LAMP1 (Invitrogen, Clone eBio1D4B, catalog #14-1071-82, lot # 2162716, 1:500 dilution), anti-alpha-synuclein (phospho S129) antibody (Abcam, clone 81 A, catalog # ab184674, lot # GR3407805-1, 1:1000 dilution), anti-Alpha-synuclein (phospho S129) antibody (Abcam, clone EP1536Y, catalog # ab51253, lot # GR3437967-8, 1:1000 dilution), anti-α-Synuclein (human) monoclonal antibody (15G7) (ENZO, clone 15G7, catalog # ALX-804-258, lot # 12071802,1:500 dilution), anti-OPA-1 (BD Transduction Laboratories, clone 18, catalog # 612606, lot # 8066752, 1:1000 dilution), anti-beta-actin (Santa Cruz Biotechnology, clone 2A3, catalog # sc-517582, lot # A1118,1:1000 dilution), anti-TOM20 (D8T4N) (Cell Signaling, clone D8T4N, catalog #42406, lot # 4,1:1000 dilution), Alexa Fluor secondary antibodies (1:500 dilution) were from Invitrogen, HRP-conjugated secondary antibodies (1:1000 dilution) were from Cell signaling technology, Biotinylated secondary antibody (1:1000 dilution) from Vector Laboratories. Ub-Lys TAMRA for biochemical USP30 assays (Almac, custom synthesis), HA-Ahx-Ahx-Ub(1-75)- VME probe for cellular ubiquitin probe binding assay (Almac, AUB-151), Human His6-USP30(57-517) (Boston Biochem, #E-582), Mouse USP30(57-517) (Boston Biochem, #E-582M), antimycin A (Sigma, #A8674), oligomycin A (Sigma, #495455), trichlorfon (Sigma, #45698). MTX115325 synthetic route and analytical methods are detailed in patent application WO2021/249909A1. Genomic DNeasy extraction kit from Qiagen, Hot Start Polymerase PCR kit purchased from EMDmillipore (catalog # TB341-KOD) and Primer pairs synthesized from Sigma Aldrich. The BLOXALL, ABC kit, DAB kit and Vectashield anti-fading mounting medium were purchased from Vector Laboratories.

Primer pair for wildtype *Usp30*: 5'-CTTGGGAAGGGATCTTGTGC-3', 5'-GTCCTCGGTGACTTCTTGGC-3'; Primer pair for mutant *Usp30*: 5'-CTTGGGAAGGGATCTTGTGC-3', 5'- TCGTGGTATCGTTATGCGCC-3';

Primer pair for GFP-mCherry-FIS1 construct: 5'- CAAAGACCCCAACGAGAAGC −3', 5'- CCCAAGGCACACAAAAAACC −3', and wildtype control: 5' −CTCTTCCCTCGTGATCTGCAACTCC- 3', 5'-CATGTCTTTAATCTACCTCGATGG −3'.

## Adeno-associated vectors (AAV) 1/2 stereotaxic injection

For knockout studies, animals at 12 weeks of age were anesthetized with ketamine/xylazine and placed in a stereotaxic frame (myNeurolab, Leica Microsystems) with a mouse adapter. 2 μl of suspended AAV vector ($1 × 10^{10}$ viral genome copies) was delivered at a speed of 50 nl/second through a pulled glass micropipette pipette (World Precision Instruments) into the right SNpc with stereotaxic coordinates

(AP: −3.0 mm, ML: −1.3 mm, DV: +4.7 mm). After a 5-min period where the pipette lies in place after dispensation of the virus, the pipette is retracted slowly.

For the pharmacological inhibitor study, approximately 10-week-old male mice were anesthetized, and placed in a stereotactic frame on a homeothermic blanket system with a core temperature maintained at 37.0 °C. A blunt injection needle (30 G) connected to a 25 μL Hamilton micro syringe mounted on a digitally guided infusion unit (Digital Lab Standard™, Harvard Apparatus) and pump (Pump 11, Elite Nanomite, Harvard Apparatus) was lowered into the level of SNc. 2 μL of AAV-A53T ($5 \times 10^{12}$ viral genomes/mL) was infused unilaterally to the right SNc at a rate of 0.2 μL/min using a micro infusion pump at the following coordinates (AP/ML relative to the bregma) AP −3.0 mm (posterior from bregma), ML 1.3 mm, DV −4.2 mm (from the brain surface). After a 5-min period where the pipette lies in place after dispensation of the virus, the pipette is retracted slowly over 1 min. For the pharmacological inhibitor study, the contralateral hemisphere was not injected with the control virus.

### MTX115325 dosing and in-life blood sampling for compound level measurements

MTX115325 was dosed twice daily at 12-h intervals for ten weeks post AAV administration (first dose administered approximately 6–8 h after viral injection) via oral gavage in 0.5 % w/v HPMC with 0.1 % v/v Tween-80, 10 mL/kg. In-life blood samples were taken at 0.5, 1, 2, 4, 6 h, timed relative to the first daily dose, from n = 3 animals per time point to give a composite profile from animals in each group. The in-life blood samples were collected from the saphenous vein (20 μL), transferred in an Eppendorf tube immediately and mixed with an equal volume (20 μL) of ice-cold Serine esterase inhibitor, trichlorfon (100 μM), the sample was frozen, stored at −80 °C and compound levels analyzed by LC-MS.

### Behavioral assessment

The cylinder test assessed spontaneous forelimb usage at 28 weeks after the AAV1/2 injection. Mice were placed into a transparent plexiglass cylinder of 12 cm diameter and 30 cm height and were video recorded for 10 min or 30 times of rearing, whichever comes first. The videos were scored post-hoc by an observer blinded to the genotype and treatment condition. Each rearing of the mice was analyzed for the number of touches of the inner surface of the cylinder with either the right (ipsilateral), the left (contralateral) or both forelimbs simultaneously. The final data were presented as a percentage of the contralateral (left) forelimb used by calculation with the equation: (contralateral forelimb +both forelimb)/(contralateral forelimb + ipsilateral forelimb + both forelimb × 2) × 100. The calculated percentage reflects the asymmetrical usage of the affected forelimb as follows: 50% = symmetric use of both forelimbs; <50% = preference of the intact (ipsilateral) forelimb; >50% = preference of the affected (contralateral) forelimb.

### Immunohistochemistry and immunofluorescence staining

For histological studies, the brain was removed after transcardial perfusion with phosphate buffered saline (PBS) followed by 4% paraformaldehyde in PBS (pH 7.0). The brains were then postfixed overnight in 4% paraformaldehyde in PBS before being immersed in 30% sucrose in PBS to cryopreserve the brains in preparation for cryostat sectioning. Immunostaining was performed on 40-μm thick free-floating coronal sections. For DAB staining, the brain sections were pretreated with BLOXALL (Vector Laboratories) for 10 min to exhaust endogenous peroxidases. The brain sections were then incubated with primary antibody at 4 °C degrees overnight. The sections were processed with secondary antibody incubation with or without an ABC kit (Vector Laboratories) and 3′-diaminobenzidine as a chromogen (DAB, Vector Laboratories) on the next day. The sections were mounted on

Superfrost Plus Microscope Slides (Fisher Scientific) and coverslipped with Vectashield antifading mounting medium (Vector Laboratories) for fluorescence imaging or dehydrated and coverslipped with Permaslip medium (Alban Scientific) for neuronal counting.

### Mitophagy puncta and colocalization analysis

For knockout studies, fluorescence images were captured with a Leica STELLARIS 5 confocal microscope equipped with a 63X objective. The mitophagy puncta were identified as mCherry only fluoresecence signal (mitophagy puncta) in Fig. 1 and colocalized OPA-1 and LAMP1 fluorescence signal in Suppl. Fig. 3c. For analysis of mitophagy puncta, the images were processed by the particle analysis function of ImageJ (NIH). For analysis of colocalization, the images were processed with the JACoP (Just Another Colocalization Plugin) plug-in of ImageJ (NIH).

For inhibitor studies, live imaging was applied to mitophagy puncta analysis in SH-SY5Y mito-QC cells. SH-SY5Y mito-QC cells were plated overnight at 7000 cells/well into black walled PhenoPlate 96-well microplates in DMEM/F-12 (phenol red free) supplemented as above. SH-SY5Y mito-QC cells were incubated with NucBlue™ live stain for 20 mins then washed. Cells were pre-treated with DMSO or 0.037 μM–1 μM MTX115325 for 20 min prior to treatment with either 0/0 μM, 0.1/0.1 μM or 1/1 μM Antimycin/Oligomycin (final DMSO conc 0.11%). Cells were live imaged using the Operetta CLS Type HH1600 (optical mode set to confocal, 40x water objective) at timepoints 0 h, 4 h, 20 h, 24 h, 48 h and 72 h. Images were acquired using the following channel settings: mCherry; excitation 530/60 nm, emission 570/650 nm, GFP; excitation 460/90 nm, emission 500/650 nm, Hoechst; excitation 355/85 nm, emission 430/500 nm. For all conditions tested, quantification of mitophagy was performed from three independent experiments (with three technical replicates capturing 11 fields of views, 6 z-stacks).

Image Quantification: For analysis of mitophagy, images were processed with the Harmony analysis software using the "SH-SY5Y-mito-QC_analysis 2" sequence. For mitolysosome identification the following analysis steps were performed: (1) Identification of the cell nuclei using Hoechst staining, (2) Identification of the cell cytoplasm using a deep phase contrast (DPC) image, (3) Identification of the mitochondrial network, using the Ser-Bright function on the mCherry channel within the cytoplasm, (4) Mitochondrial network partitioning using the FindSpots function, (5) Identification of mitophagic puncta with the partitioned mitochondrial network, using an mCherry:GFP ratio threshold of mean >+3 standard deviations from the 24 h DMSO only treated cells.

### Western blot analysis for knockout studies

Brain tissue samples were homogenized in ice-cold radio-immunoprecipitation (RIPA) lysis buffer (50 mM Tris-HCl, pH 7.4, 150 mM NaCl, 0.1% SDS, 1% NP-40, 2 mM EDTA, 1 mM DTT, 1 mM PMSF, 200 μM Na3VO$_4$, 50 mM NaF) and protease inhibitors (Roche) on ice for 20 min with brief sonication, and centrifuged at 15,600 × $g$ for 15 min. The supernatants were collected for immunoblot analysis. The protein concentration of samples was determined with a BCA protein assay kit (Thermo Fisher Scientific). Denatured samples were heated to 70 °C in NuPAGE LDS sample buffer (Invitrogen) and loaded onto NuPAGE Bis-Tris gels (Invitrogen) for protein separation. Following electrophoresis, proteins were transferred from the gel to a nitrocellulose sheet by electrophoresis. The sheet was washed with deionized water and incubated overnight at 4 °C with a primary antibody (see antibody sources in the second paragraph of the Materials and Methods section). The sheet was then incubated in the presence of HRP-conjugated secondary antibodies on the next day, and immunoblots were visualized using an enhanced chemiluminescence kit (Thermo Fisher Scientific). All the representative blots used in the figures are presented in Source Data file.

## Neuronal Counting and densitometry

For neuronal counting in the knockout study, the midbrain of each mouse was sectioned using a Leica cryostat machine into six series of 40 μm coronal sections and one series was stained with anti-TH antibodies for dopaminergic (TH + ) neuronal counting, respectively. The serial brain sections were then scanned with NanoZoomer XR Digital slide scanner (Hamamatsu) for neuronal counting with the QuPath v0.2.0 software (https://qupath.github.io) as previously described[49].

For striatal densitometry in the knockout study, TH-stained sections were imaged using a light microscope fitted with a camera (SPOT). Images were captured using a 10 × objective and fixed exposure settings. Densitometry was performed to quantify the intensity of TH immunostaining in the striatum ipsilateral to the stereotaxic injection and in the contralateral striatum. The average intensity of TH staining in a fixed region of the striatum was quantified with ImageJ software (NIH). The relative optical density of TH+ fibers was normalized by subtracting the background intensity of the cortex, and the percentage of ipsilateral over contralateral relative density was calculated and used for statistical analysis.

For the compound study, the fixed, cryoprotected and frozen midbrain samples were sectioned as coronal sections at 100 μm intervals through SNpc and ventral tegmental area (VTA) and mounted on glass slides. Sectioning was started at −2.6 mm from bregma and continued to −3.7 mm from bregma in the AP axis (coordinates based on Franklin & Paxinos: The Mouse Brain in Stereotaxic Coordinates). For the immunostaining, the sections were stained with anti-TH following permeabilization. The stained sections were scanned with the Olympus VS120 slide scanner and analyzed with a VIS-Visiopharm Integrator System (Visiopharm, Denmark) by a blinded data analyst. All cells of interest with visible nuclei were labeled manually and counted using an APP (Analysis Protocol Package), scaled by section thickness and sectioning interval to estimate the total cell population in the SNpc. For immunostaining of Ser129 αSyn phosphorylation, TSA amplification was used with primary antibody ab51253 (Abcam). Blinded analysis was used, and positive cells per area were determined. For dual IF IHC of Iba1 (WAKO) and GFAP (ab4674 – Abcam), the stained sections were scanned and analyzed by a blinded data analyst. Analysis determines the percentage of positive staining area [%area = (stained area/region area) * (100)]. To calculate % of stained area, the same threshold value was applied to all sections with the same antibody staining. All pixels above the threshold value were considered positive. All striatal densitometry and neuronal counting data for both KO and inhibitor studies were obtained by an investigator blinded to genotype and treatment.

## Measurement of dopamine and dopamine metabolites

For the knockout studies, mice were sacrificed at 28 weeks post-injection. The brains were rapidly removed, placed into a chilled brain matrix, and sliced into 1 mm thick coronal sections on ice. The sections were then placed into ice-cold saline. The striatum was dissected from these 1 mm sections, snap-frozen and used for HPLC analysis of dopamine and its metabolites at the Neurochemistry Core, School of Medicine, Vanderbilt University. Levels of dopamine and its metabolites were normalized to ng/mg of protein input and statistically analyzed with GraphPad Prism software. For the compound dosing study, mice were sacrificed at ten weeks post-injection. Snap-frozen tissue samples were homogenized, the analytes were separated by HPLC on a Kinetex EVO C-18 reversed-phase column, and the levels of analytes were calculated using external standards and expressed as ng/g wet tissue.

## USP30 biochemical fluorescence polarization assays

USP30 enzyme was diluted in reaction buffer (40 mM Tris HCl, pH 7.5, 0.005% Tween 20, 0.5 mg/ml BSA, 5 mM BME) to achieve a final concentration of 8 nM for human USP30 and 10 nM for mouse USP30, the compound was added and incubated for 30 min at room temp then reactions initiated by addition of Ub-Lys-TAMRA (final conc of 50 nM). Fluorescence was measured immediately after the addition of substrate and following a 2 h incubation on Pherastar Plus or FSX with λ Excitation 540 nm and λ Emission 590 nm.

## Cell USP30 ubiquitin probe binding assay

Hela cells stably overexpressing YFP-Parkin were treated with appropriate concentrations of test compound or vehicle (DMSO) control for 1 h at 37 °C. Whole-cell lysates were prepared by scraping the cells into cold PBS, centrifuging and lysing in lysis buffer (50 mM Tris-base, pH 7.5, 50 mM sodium chloride, 1% NP-40/Igepal CA-630, 2 mM MgCl2, 10% Glycerol, 5 mM β-mercaptoethanol, cOmplete mini tablets EDTA free, PhosStop tablets) cleared cell lysate, was incubated with a final concentration of 2.5 μM HA-Ahx-Ahx-Ub-VME probe for 15 mins at room temperature. The reaction was stopped by the addition of 5x SDS sample loading buffer and boiling for 5 min at 95 °C. Proteins were separated on NUPAGE 4–12% Bis-Tris Gel transferred to nitrocellulose and USP30 was detected using anti-USP30 Sheep antibody and a rabbit anti-sheep secondary and visualized using ECL reagent on a GE LAS4000 imager. Target engagement was measured by quantitation of the bands corresponding to USP30 and USP30 bound to the Ub-VME probe and expression of this proportion compared to vehicle-treated control.

## Cell TOM20 ubiquitination assay

Hela cells overexpressing YFP-*Parkin* were treated with 1 μM Antimycin A and 1 μM Oligomycin A (AA/OA) and test compound or DMSO for 90 mins. Following treatment, media was removed, and cells were washed in DPBS and then lysed in NP40 lysis buffer on ice. Proteins were separated on a NUPAGE 4–12% Bis-Tris Gel, transferred to a nitrocellulose membrane, and TOM20 was detected using an anti-TOM20 antibody and a goat anti-rabbit secondary and visualized using ECL reagent on a GE LAS4000 imager. The bands corresponding to all forms of ubiquitylated TOM20 were quantified and normalized to the same bands for AA/OA treated cells. The point where the signal is 1.5x the AA/OA control (EC1.5x) and the EC50 were measured.

## Mouse pharmacokinetics and brain CETSA analysis

Mouse clearance was calculated from blood compound profiles following IV administration of 2 mg/kg MTX115325 and sequential tail vein blood sampling, and mouse oral bioavailability was calculated from compound profiles following oral dosing by gavage of 10 mg/kg MTX115325 and sequential tail vein blood sampling. Mouse Kpu,u estimates were calculated from blood and prefrontal cortex compound profiles following oral dosing of 10 or 30 mg/kg by gavage, with sequential blood sampling from the jugular vein and sequential brain interstitial fluid sampling via a microdialysis probe in the prefrontal cortex. Measured drug levels from the PFC were considered to be free, whereas blood compound levels were multiplied by the blood-free fraction of MTX115325 to calculate unbound drug levels.

Male C57BL/6 J mice aged 6-8 weeks were treated orally with either vehicle (0.1% Tween-80/0.5% HPMC) or 10 mg/kg PO MTX115325 in a volume of 10 ml/kg. Animals were sacrificed by overdose of isoflurane followed by cervical dislocation 15 min, 30 min, 1 h, 2 h, 4 h, 8 h, or 24 h post dosing followed by transcardial perfusion with ice cold PBS/10% EDTA. Whole blood and cortex were harvested and frozen on dry ice. Samples were stored at −80 °C until processing.

Target engagement was measured by cellular thermoshift assay CETSA® assay technology (under license from Pelago Biosciences, Sweden). Brain cortex samples were subjected to heat shock by heating at 46 °C for 8 min in Mg⁺ and Ca²⁺ free PBS with PIC, Phospho-stop and 1 mM PMSF. Immediately after heat shock, samples were snap frozen in liquid nitrogen. Samples were thawed on ice homogenised for 2 × 45 s in a Retsch MM 400 tissue homogeniser. 0.4% NP40 (final concentration) was added to the lysates before subjecting them to

three freeze thaw cycles. Aliquots of samples were removed for BCA protein concentration determination before samples were centrifuged at 17,000 × g for 20 min and the supernatant stored at −80 ℃ until Western Blotting. Membranes were washed with TBS-T then blocked with 10% milk TBS-T for 30 min before the addition of the USP30 antibody (sheep polyclonal anti-USP30 (University of Dundee) 1:500 in TBS-T with 5% milk) and incubation at 4 ℃ on a shaker for 2 nights. On the final day, the USP30 antibody was incubated at room temperature on blot the for 1 h before washing. ECL detection reagent was used for 5 min before developing on a AI600 LAS imager. Where samples were run on several gels, benchmarking samples were used with the same sample run on each gel. USP30 stabilisation was determined by western blot quantification, normalised to the benchmarking sample then to HSC70 as a loading control. Data are presented as averages for each group. Absolute USP30 and fold change to the vehicle average were calculated. USP30 target engagement was reported as a percentage range, where the vehicle dosed group average was equal to 0% and the group with maximum average target engagement fold change above vehicle was equal to 100%. Percentage range calculated: (x-min)/(max-min)*100.

## Statistical analysis

The statistical analysis was conducted in Prism 9 (GraphPad) and R. Significant differences between the two groups were performed with a two-tailed, unpaired equal variance Student's $t$ test unless. Mann–Whitney U non-parametric tests were used when the assumptions of the $t$ test were not met. $P < 0.05$ was considered a statistically significant difference. $*P < 0.05$, $**P < 0.01$, $***P < 0.001$, and $****P < 0.0001$. The value of $n$ per group is indicated within each figure legend. The statistical analysis pipeline phenotypic data presented in Supplementary Data 1 and Supplementary Fig. 1 as well as data from Supplementary Fig. 4 were performed in R.

## Reporting summary

Further information on research design is available in the Nature Portfolio Reporting Summary linked to this article.

## Data availability

All data are in the main text or the supplementary materials. Source data are provided as a Source Data file with this paper. Source data are provided with this paper.

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

## Acknowledgements

We thank Dr Laura Parton for project management/co-leadership of MTX115325, providing the foundation for Mission compound studies and MTX115325 development in PD. We also thank Drs. Krutika Joshi and Veronique VanderHorst of BIDMC for their support with slides scanning for TH+ neuronal counting and TH+ fiber densitometry. We thank Mat-thew Jacobsen for the pathology assessment of mouse livers and Nirav Prakas Patel for early work on mouse ESCs. We would like to thank Prof. Maria Grazia Spillantini for her valuable comments on the manuscript. Research in the D.K.S. lab is funded by an NINDS grant (R21NS109408), the Weston Brain Institute and the Owens Foundation. Research in the G.B. lab is supported by the UK Dementia Research Institute, that receives contributions from UK DRI Ltd, the UK MRC, the Alzheimer's Society, and Alzheimer's Research UK as well as a grant from the Romanian Ministry of Research, Innovation and Digitization (no. PNRR-III-C9-2022-I8-66; contract 760114). IGG was funded by a grant from the Medical Research Council, UK (MC_UU_00018/2). Research in the S.P.J. lab is funded by Cancer Research UK Discovery grant (DRCPGM \100005), and ERC Synergy grant DDREAMM (855741). This project has received funding to SJ from CRUK programme grant C6/A11224, C6/A18796 and Wellcome Investigator Award (206388/Z/17/Z) together with core infrastructure funding by Cancer Research UK (C6946/A24843) and Wellcome (WT203144). The Sanger Mouse Genetics Pro-ject was supported by the Wellcome Trust (098051). CRUK sup-ports D.J.A.

## Author contributions

D.K.S., G.B., S.J. and P.T. acquired the funding and supervised the pro-ject; D.K.S. and T.-S.Z.F. conceived and designed the αSyn genetic study. T.-S.Z.F. performed the experiments and data analysis for the genetic study. G.B., D.J.A. and S.J. created and phenotyped the Usp30 KO mice with help from C.J.L. and Y.S.; N.A.K. performed the statistical analysis of the Usp30 KO mouse pipeline phenotyping; I.G.G. and J.F.Z. crossed Usp30 KO with mito-QC mice and carried out peripheral mito-phagy analyses. A.C.P. and P.T. led the drug study and created all data related to USP30 inhibitor work with help from M.K., C.A.L., R.W., R.M., S.A., L.B. and N.M. G.B. T.-S.Z.F., S.E. P.T. and D.K.S. reviewed and edited the manuscript. All authors approved the manuscript.

## Competing interests

A.C.P., M.K., C.A.L., R.W., R.M., S.A., L.B. and N.M. and P.T. are employees of, or former employees of, Mission therapeutics. S.J. is a founder, shareholder and board member of Mission Therapeutics. All other authors declare no competing interests.
