## [Peer Review File · Nature Communications]

Knockout or inhibition of USP30 protects dopaminergic neurons in a Parkinson's disease mouse modelREVIEWER COMMENTS

Reviewer #1 (Remarks to the Author):

USP30 has been proposed as an actionable target for Parkinson's disease and several other indications, based on its ability to suppress mitophagy. Here the authors combine characterisation of a USP30^{-/-} mouse model and a novel cyanopyrrolidine inhibitor with drug like properties applied to a mouse model of PD. They also provide in vivo data for the proposed role in mitophagy using a mouse reporter model (MITO-QC). Most of these observations recapitulate findings by others in vitro. The headline novel result is that in a mouse model for PD that involves over-expression of α -synuclein, loss of USP30 and drug treatment both display some rather remarkable remedial properties. This is highly interesting and likely to receive a lot of attention and potentially large sums of money being invested. It requires serious scrutiny to ensure that it does more good than harm to the field. It is not clear that this is the expected result based on the current conceptual framework, must it necessarily reflect mitophagy? The key thing that is missing is any mechanistic insight into the effect. Explanations offered are merely hand waving and sometimes counter intuitive. There is a part of me that thinks the manuscript would be clearer if the mitoQC elements are left out, because they are not clearly linked to the main claim.

Figure 1g: This looks at quantitation of the mitochondrial marker mitoQC with a lysosomal marker LAMP2. Number of mice used is given, but not number of neurons quantitated per mouse. I am not sure why this assay for mitophagy should be better than the normal assay employed which takes advantage of mitoQC spectral properties changing in a low pH environment. They supposedly adopt this approach to alleviate concern over the fact that the mitoQC is also expected to label peroxisomes. Contrary to the authors claim the new assay does not help with that discrimination and in theory at least any overlapping punctae can just as easily be peroxisomes. USP30 has been linked to pexophagy as well as mitophagy (doi: 10.15252/embr.201745595,doi: 10.26508/lsa.202101287, DOI: 10.1083/jcb.201804172).

I am also left wondering if this mitophagy enhancement effect is specific to the SNpc or whether it is seen elsewhere.

Figure 2 and 3 make a very strong case for the significance of their finding for the remedial effects of USP30 loss in the α -synuclein model. However, despite the inclusion of mito-QC mouse data, we are not given any insight into how the mitophagy parameter changes in response to the α -synuclein expression, nor the effect of USP30. I think this is a major omission. I am also wondering why the basal TH staining looks higher in the KO animals than the controls - is this a robust finding? It should at least be commented upon.

Figure 4: introduces a novel cyanopyrrolidine compound related to the same chemical series reported in previous papers (doi: 10.1016/j.molcel.2020.02.01, doi: 10.26508/lsa.202000768, DOI: 10.1042/BCJ20210508). It appears that this compound may have better drug like properties in terms of selectivity compared with these preceding studies, which also used Ub-TOMM20 as a biomarker. The figure also provides some useful pharmacokinetic data. Reference 70 describes an alternative series of highly specific USP30 inhibitors, that have since been better characterised than given credit for (DOI: 10.26508/lsa.202101287). Are there any pros and cons for one or the other scaffold going forwards to the clinic that the authors would like to articulate?

Figure 5 5C/D/E - Stats should compare vehicle compared to inhibitor. Non-significance between EV and SNCA does not prove a lack of difference.

Data for table 1 could be included in the supplementary material.

Supplementary data: changes in liver fat composition have been previously reported for USP30^{-/-} mice and this should be referenced DOI: 10.1002/hep.31249. Are there any effects of the drug on liver fat- this would be interesting. I would also like to see some discussion of the implications of this paper DOI: 10.1126/science.abe9977. Do any of the new mouse data speak to the potential

issue of cytotoxic T cell function.

The text contains some problematic statements, examples of which are below

"We show that in vivo USP30 inhibition represents a viable strategy for enhancing the clearance of damaged mitochondria through mitophagy by counteracting PARKIN-related ubiquitination of the outer mitochondrial membrane proteins (like TOM20) in an α Syn based mouse PD model." - most of this statement is unproven. They don't show mitophagy in the α syn model and don't show any Parkin dependence in any setting. The only relation to Parkin is the blot of TOM20 with inhibitor in Parkin-Hela cells, but they don't show the same without Parkin.

"We used a novel strategy for upregulating mitophagy by Usp30 deletion, with similar results obtained with a pharmacological USP30 inhibitor. These strategies to reduce USP30 lead to enhanced mitophagy and potent protection against α Syn toxicity." - Again they don't show their inhibitor enhances mitophagy, they only show enhanced ub-TOM20 with overexpressed Parkin. If they want to make a strong point about the inhibitor enhancing mitophagy they should demonstrate this (with endogenous Parkin). This should be a straight-forward ask.

"These data collectively demonstrate potent protection against α Syn toxicity by enhancing mitophagy through the loss of USP30." - Again, they don't show the effect is via mitophagy.

"To determine if upregulation of mitophagy in Usp30 KO mice injected with AAV-SNCA is associated with decreased α Syn pathology" - confusing, they don't show mitophagy in KOs injected with AAV-SNCA.

Reviewer #2 (Remarks to the Author):

This manuscript outlines the generation of a USP30 knockout mouse model with subsequent phenotyping, along with the investigation of the interaction between USP30 knockout and SNCA overexpression. The authors then add to the validity of the knockout results with work using a USP30 inhibitor. The topic area of this manuscript is one of great interest to the field of Parkinson's research particularly as well as general neurodegeneration and hence the work in this manuscript is of high interest to a large scientific community.

The authors have carried out extensive characterisation of the USP30 k/o mouse they have generated including with ageing, this information is required if USP30 is to be a viable therapeutic target for Parkinson's. It seems odd however that the authors do not compare their results with those gained by Phu et al, 2020 who also generated a USP30 k/o mouse model (although with different characterisation). It would strengthen the manuscript greatly if the authors were to include a comparison of the results of this USP30 k/o and the one previously published. The fact the authors have described a potential phenotype with age in the USP30 knockout mice which is beneficial is worthy of further studies which include mechanistic work. This would strengthen the case that those beneficial effects of USP30 k/o with age are related to the deubiquitinase function of USP30 and not some other mechanism. Have the authors investigated the USP30 k/o / mitoKeima mouse during ageing in those tissues specifically protected from age related disease? The authors validate work published by others that USP30 k/o increases mitophagy rates and have specifically investigated this in SN neurons, although the most relevant cell type to Parkinson's Disease, from a therapeutic perspective, it would be very beneficial for the authors to have investigated mitophagy rates in other tissues and even cell types within the brain. It is important to understand if the USP30 k/o effect is also present in glial cells of the SN and if the whole body effects are due to the mechanistic changes in mitophagy rates.

The study the authors carry out in the USP30 k/o animals with AAV-SNCA injection is very interesting and brings together multiple key pathways in Parkinson's; this is also novel for the study. Overall, the results are clear and well articulated. With respect to the TH neurons after AAV-SNCA injection, there appears to be a difference (quite large) between the effect this has in the WT mice versus the mitoKeima mice, however the authors have not commented on this. Could they please comment on this difference, why this could be and if this affects the resulting

protective effects of USP30 k/o? Furthermore, the age at which the mice were injected with AAV-SNCA is of interest with the change in age related phenotypes, it would be interesting for the authors to comment on this. The authors show clearly the 2 genders in these studies, do the authors find any differences in response to AAV-SNCA in the USP30 k/o between genders? The authors then go on to validate their findings of the USP30 k/o with using a USP30 inhibitor and show the inhibition selectivity for this compound and in vivo characteristics. This is an important piece of validation work. The data shown on the inhibitor in Figure 4 is somewhat limited, at the very least it would be essential for the authors to add in detail on the robustness of the inhibition shown in Figure 4b. In addition, could the authors show the full panel of DUB and cathepsin inhibition rather than just 2 in Table 1?

Could the authors comment on why only female mice were used for the inhibitor study? In Figure 5c-e it appears as though the USP30 inhibitor treatment has reduced the dopamine, DOPAC and HVA levels in the EV treated mice. Could the authors comment on this and if they tested this statistically? In addition, did the authors test if the AAV-SNCA is significantly different to the AAV-SNCA + USP30 inhibitor group?

The authors test 2 different doses of the inhibitor, however little dose dependent effect is seen with these doses, could the authors comment on any dose dependent effect and if not, what type of study would be required to understand this?

The authors have shown blood levels of the inhibitor, however, the inhibitor treated dataset and study is hampered in usefulness as no target engagement work has been done, including mitophagy rates, ubiquitination assays etc. This makes the full interpretation of the USP30 inhibitor study difficult. Are there any differences between the effectiveness of the USP30 inhibitor in vivo in different cell types or tissues? These are crucial points to be addressed if the compound and target are to be progressed to the clinic.

In the discussion, the authors miss discussing their findings in relation to much of the current literature around USP30 and USP30 inhibitors, adding a fuller discussion of all of the USP30 inhibitor studies would greatly enhance the manuscript.

Finally, the two study designs are different, in that the USP30 k/o study investigates protection of USP30 as the k/o is present before the AAV-SNCA pathology begins, whereas in the inhibitor experiments the AAV-SNCA is given first before inhibitor dosing begins. It would be useful if the authors could present data and/or comment on these two paradigms and what the results and effects could tell us about USP30 as a therapeutic target for PD.

A couple of minor points:

In the abstract there is a typo in line 3, it should read mitochondrial

In the introduction line 4, the authors needs to add AR-PD

The last line of page 3 has too many siRNA

The title of Supplementary figure 4 does not read correctly

Reviewer #3 (Remarks to the Author):

In this manuscript, Fang et al. describe a protective role for USP30 in counteracting the toxicity of a-synuclein (aSyn) in dopaminergic neurons. USP30 KO mice are protected in the AAV-aSyn overexpression model, and display reduced phospho-aSyn pathology and increased mitophagy in dopaminergic neurons. Pharmacological inhibition of USP30 produces similar benefits in mice. The significance of the mitophagy pathway for neurodegeneration in vivo is still debated and this manuscript builds on previous literature by showing that boosting mitophagy could provide protection against aSyn stress in vivo. Additional controls and experiments outlined below should strengthen the paper for publication in Nature Communications.

Fig 1f - In the mitoQC images shown, the most noticeable change is an overall increase in the reporter signal (both red and green) as opposed to the appearance of "mCherry/red only" punta. It will be better to highlight the mitophagy puncta counted in the images shown and provide

zoomed-in images.

The decrease in phospho-aSyn signal in USP30 KO (despite the preservation of dopaminergic neurons) mice is striking. The authors should provide more mechanistic insights into this observation. Is aSyn expressed to the same level between the genotypes (or is the total level of aSyn also reduced)? pSyn staining at an earlier time point should also inform if pSyn pathology forms normally but cleared faster in the KOs.

Is the baseline TH terminal intensity in the striatum different across groups in Figure 3A (+Ev image in Mito-QC/USP30 KO seems higher compared to the wild-type genotypes)?

The authors could discuss that intermittent dosing may be sufficient for efficacy given that the target coverage was between 7.5-12 hours daily.

Fig S4b - Around half of the AAV-aSyn injected animals did not develop the pSyn pathology even in the Vehicle dosed animals. What is the reason for this? Were injections QCed by following the needle track for proper access to the nigra?

REVIEWER COMMENTS

Reviewer #1 (Remarks to the Author):

1. *"The headline novel result is that in a mouse model for PD that involves over-expression of a-synuclein, loss of USP30 and drug treatment both display some rather remarkable remedial properties. This is highly interesting and likely to receive a lot of attention and potentially large sums of money being invested. It requires serious scrutiny to ensure that it does more good than harm to the field."*

We thank the reviewer for the general constructive comments and suggestions and the observation that our work shows that USP30 loss or inhibition "displays some rather remarkable remedial properties. This is highly interesting and likely to receive a lot of attention [...]".

2. *"It is not clear that this is the expected result based on the current conceptual framework, must it necessarily reflect mitophagy? The key thing that is missing is any mechanistic insight into the effect. Explanations offered are merely hand waving and sometimes counter intuitive. There is a part of me that thinks the manuscript would be clearer if the mitoQC elements are left out, because they are not clearly linked to the main claim."*

We agree with the reviewer that, at the moment, both our work presented here, as well as work by others, does not fully clarify the molecular mechanism. We can say that upon USP30 knock-out or inhibition we see a specific increase in mitophagy that correlates with improved neuronal survival and rescue of pathophysiology associated with this mouse model. We think that the mitophagy data and discussion represent an important part of the story, particularly given the previously published data indicating an important role of mitophagy in PD. It would seem to be a substantial gap to discuss the impact of targeting USP30 without discussing the effect on mitophagy. We agree with the important point that we have to clearly indicate that we have not proven if mitophagy is the cause of the neuroprotection that we see from targeting USP30. We have now adjusting the wording in the manuscript accordingly to be clearer on this point. In addition, as the reviewer will see we added additional data as requested by other reviewers on the role of USP30 on mitophagy in other models (human neurons).

3. *"Figure 1g: This looks at quantitation of the mitochondrial marker mitoQC with a lysosomal marker LAMP2. Number of mice used is given, but not number of neurons quantitated per mouse. I am not sure why this assay for mitophagy should be better than the normal assay employed which takes advantage of mitoQC spectral properties changing in a low pH environment. They supposedly adopt this approach to alleviate concern over the fact that the mitoQC is also expected to label peroxisomes. Contrary to the authors claim the new assay does not help with that discrimination and in theory at least any overlapping punctae can just as easily be peroxisomes. USP30 has been linked to pexophagy as well as mitophagy (doi: 10.15252/embr.201745595,doi: 10.26508/lsa.202101287, DOI: 10.1083/jcb.201804172).*

We analysed the mCherry and LAMP1 colocalisation puncta to avoid the low fluorescent intensity of GFP in the mito-QC mice in the brain. We have now shown the same is the case with only looking at mCherry (Fig. 1g). We quantified 10 DA neurons per mouse for the average number of mitophagy puncta in the mice, and we have now included this information in the figure legends. To make sure the mito-QC expression on the peroxisome is not an issue, we also did an OPA-1 and LAMP1 co-localization study to assess the mitophagy puncta in SNpc DA neurons in the AAV-SNCA-based PD model. The data is presented in Figure S4a and it shows the same pattern where Usp30 KO mice have increased mitophagy.

4. *"I am also left wondering if this mitophagy enhancement effect is specific to the SNpc or whether it is seen elsewhere."*

This is an important point; we have now quantified mitophagy in other parts of the brain (cortex, hippocampus) and other peripheral tissues such as the muscle and these data are now presented in Figure S3a and show now that USP30 KO mice show no difference in mitophagy in the other tissues. We also quantified mitophagy in other parts of the brain (cortex and hippocampus) and show that Usp30 KO mice have higher mitophagy levels in these areas too (Suppl. Fig. 3 c,d). This is consistent with data generated ex vivo in Usp30KO hippocampal neurons published by Phu et al 2020

5. *"Figure 2 and 3 make a very strong case for the significance of their finding for the remedial effects of USP30 loss in the a-synuclein model. However, despite the inclusion of mito-QC mouse data, we are not given any insight into how the mitophagy parameter changes in response to the a-synuclein expression, nor the effect of USP30. I think this is a major omission."*

We thank the reviewer for this very important point. We have now analysed the mitophagy in the SNpc DA neurons of the a-synuclein model comparing USP30 WT mito-QC to USP30 KO mitoQC and show that the level of

mitophagy at the timepoint assessed (28 wks post viral delivery) is not affected by the delivery of a-synuclein and USP30 KO leads to increased mitophagy independent of presence or absence of a-synuclein (Figure S4a,b). Our interpretation is that a-synuclein is probably in a different pathway (non-epistatic) than USP30 and this data fits with a model where mitophagy increase can rescue the DA neurons affected by a-synuclein accumulation (with the caveat that we don't know what happened to the already dead DA neurons).

6. *I am also wondering why the basal TH staining looks higher in the KO animals than the controls - is this a robust finding? It should at least be commented upon.*

We were also puzzled initially by the higher TH staining, so we performed densitometry comparing WT + AAV-Ev as compared to WT + AAV-A53T-SNCA and we noticed that when we injected the AAV-53T-SNCA on the Ipsilateral side the Contralateral side is also affected (see same results from another group, *Behav Brain Res.* 2022 Jul 5;429:113887. doi: 10.1016/j.bbr.2022.113887. Epub 2022 Apr 8. PMID: 35405174); the same pattern can be observed in the Mito-QC mice as well as MitoQC-USP30 mice. We present this data, including the quantification and statistics, in Figure S4 c,d. Since we got the reviews back, we also asked other colleagues that use this model and they all said that they observe this, thus we think it must be a general effect that should be perhaps better accentuated in the literature; we have now added also a small section in the materials and methods on this.

7. *“Figure 4: introduces a novel cyanopyrrolidine compound related to the same chemical series reported in previous papers (doi: 10.1016/j.molcel.2020.02.01, doi: 10.26508/lsa.202000768, DOI: 10.1042/BCJ20210508). It appears that this compound may have better drug like properties in terms of selectivity compared with these preceding studies, which also used Ub-TOMM20 as a biomarker. The figure also provides some useful pharmacokinetic data. Reference 70 describes an alternative series of highly specific USP30 inhibitors, that have since been better characterised than given credit for (DOI: 10.26508/lsa.202101287). Are there any pros and cons for one or the other scaffold going forwards to the clinic that the authors would like to articulate?”*

We appreciate that this has been recognised. However, clear articulation of the chemistry and design process that leads to pharmaceutical property improvement is beyond the scope of our publication. Regarding comparison to the alternative series cited, again, we would refrain from making any explicit statements about why our series might show better pharmaceutical properties as it would be highly speculative. By providing the structure of MTX115325 we are being as open as possible at this stage of development and the relevant researchers/chemist can compare their series to our compound.

8. *“Figure 5 5C/D/E - Stats should compare vehicle compared to inhibitor. Non-significance between EV and SNCA does not prove a lack of difference.”*

We thank the reviewer for the suggestion. Because the variability associated with injection, and individual animal response to the A53T alpha-synuclein, a very reasonable comparison would be within each mouse (i.e. non-injected NI vs A53T - back to WT; we are sorry for the typo in the initial manuscript where the NI was labelled Ev). Moreover, we always prefer to use actual quantitative raw values (ng/g). But we appreciate the point made. To ease the interpretation of the model and reduce variability, we have now provided a normalised version of the data to each group's NI side and then compared it across the groups. The data is now presented in Fig. 5 c/d/e and it shows that upon USP30i treatment Dopamine, DOPAC and HVA behave similarly with a push towards WT normal levels at 15mg/kg and borderline significance numbers, and significantly different to the Vehicle treatment at 50mg/kg USP30i treatment.

“Data for table 1 could be included in the supplementary material.”

We now added the table in Fig. S6a.

9. *“Supplementary data: changes in liver fat composition have been previously reported for USP30-/- mice and this should be referenced DOI: 10.1002/hep.31249. Are there any effects of the drug on liver fat- this would be interesting. I would also like to see some discussion of the implications of this paper DOI: 10.1126/science.abe9977. Do any of the new mouse data speak to the potential issue of cytotoxic T cell function.”*

We thank the reviewer for the important suggestion; we have now inserted the reference to the Gu et al. *Hepatology* paper and briefly discussed it in the text. This is an important observation that fits with what Gu et al. presented and thought that while it is a bit disconnected from the PD-targeted story is an important observation that is independently repeated, and people should look more into this aspect.

On the Lisci et al. *Science* paper: we didn't cite it as the material used in this paper (for initially screening T cell function and the mouse line) was provided by us, but the investigators used the mice without correctly referencing

where they got the mice from nor did they announced us when they published the paper with the material we provided. We can comment that in non-challenged conditions, we don't see an effect of the T-cell function presented on mouse health (we have kept the mice on the shelf for a long time, and we cannot distinguish them from WT in any form). Since the mice are the same line we used in our study, it must hold true what the investigators presented that "Mice lacking the deubiquitinase USP30 have Cytotoxic T lymphocytes acutely depleted of mitochondria, and these cells have reduced killing ability but normal motility, signalling, and secretion." Because of the way the investigators of this study used our provided material, we will ask for advice from the editor on what would be the best way to reference the paper.

10. "The text contains some problematic statements, examples of which are below:

a. "We show that *in vivo* USP30 inhibition represents a viable strategy for enhancing the clearance of damaged mitochondria through mitophagy by counteracting PARKIN-related ubiquitination of the outer mitochondrial membrane proteins (like TOM20) in an α Syn based mouse PD model." - most of this statement is unproven. They don't show mitophagy in the *asyn* model and don't show any Parkin dependence in any setting. The only relation to Parkin is the blot of TOM20 with inhibitor in Parkin-Hela cells, but they don't show the same without Parkin.

This is an important point and we fully agree with the reviewer that the wording of our original statement in the paper was too strong, and implied that we our data proves this mechanism whereas, in fact it had not. Our data does show that USP30 loss or inhibition leads to upregulation of mitophagy and protection against alpha-synuclein toxicity and given the clear role of mitophagy in PD pathophysiology, we suspect that these 2 are related. However, our data do not definitively prove that the upregulation of mitophagy is due to counteracting PARKIN effects or that this impact on mitophagy is the mechanism by which USP30 inhibition protects. We have now adjusted the wording in the manuscript accordingly to be more accurate on these points. For example, at the start of the discussion, we removed the phrase "by counteracting PARKIN-related ubiquitination of the outer mitochondrial member proteins".

11. "We used a novel strategy for upregulating mitophagy by *Usp30* deletion, with similar results obtained with a pharmacological USP30 inhibitor. These strategies to reduce USP30 lead to enhanced mitophagy and potent protection against α Syn toxicity." - Again they don't show their inhibitor enhances mitophagy, they only show enhanced ub-TOM20 with overexpressed Parkin. If they want to make a strong point about the inhibitor enhancing mitophagy they should demonstrate this (with endogenous Parkin). This should be a straight-forward ask."

"These data collectively demonstrate potent protection against α Syn toxicity by enhancing mitophagy through the loss of USP30." - Again, they don't show the effect is via mitophagy.

We thank the reviewer for the important observation and suggestion. In Fig. 4, we have added data generated in SH-SY5Y neurons that stably express the Mito-QC construct (Fig. 4e,f) and with normal parkin expression. These new data show that upon challenge with 0.1 μ M Antimycin Oligomycin (A/O) the USP30 inhibitor MTX115325 significantly potentiates mitophagy induction. In addition, to show that these data are relevant for human cells with endogenous Parkin, we have also performed experiments in PD-derived iPS cells (the A53T line) and show that the USP30i similarly promotes TOM20-Ub in this human model (Fig. S.)

We note that we did the Mito-QC experiments in the SH-SY5Y because of the difficulty of creating MitoQC-iPS cells stably expressing the construct.

12. "To determine if upregulation of mitophagy in *Usp30* KO mice injected with AAV-SNCA is associated with decreased α Syn pathology" - confusing, they don't show mitophagy in KOs injected with AAV-SNCA.

We have now addressed this as above (point 5) by adding these data Figure S4a,b.

Reviewer #2 (Remarks to the Author):

1. This manuscript outlines the generation of a USP30 knockout mouse model with subsequent phenotyping, along with the investigation of the interaction between USP30 knockout and SNCA overexpression. The authors then add to the validity of the knockout results with work using a USP30 inhibitor. The topic area of this manuscript is one of great interest to the field of Parkinson's research particularly as well as general neurodegeneration and hence the work in this manuscript is of high interest to a large scientific community.

We are grateful for the positive feedback and encouraging comments regarding the significance of our work to the field of Parkinson's research and general neurodegeneration. We have carefully considered your insightful suggestions and have made the necessary revisions to improve the manuscript's quality. Below, we address each of your specific points.

2. The authors have carried out extensive characterisation of the USP30 k/o mouse they have generated including with ageing, this information is required if USP30 is to be a viable therapeutic target for Parkinson's. It seems odd however that the authors do not compare their results with those gained by Phu et al, 2020 who also generated a USP30 k/o mouse model (although with different characterisation). It would strengthen the manuscript greatly if the authors included a comparison of the results of this USP30 k/o and the one previously published.

We thank the reviewer for the helpful comment. The Phu et al. generated a constitutive knockout (KO) mouse model by deleting exon 2 of the USP30 gene and investigated its impact on the mitochondrial proteome. The USP30 KO mice showed a reduced abundance of certain TOM complex subunits and March5 in mitochondria. Additionally, USP30 depletion accelerated mitophagy and led to increased basal respiration but reduced reserve capacity in hippocampal neurons. Similarly to what we observe, their findings indicate that USP30 plays a crucial role in regulating mitochondrial homeostasis, specifically in the turnover of the certain TOM complex components, and its absence influences mitochondrial metabolism and mitophagy in neurons. That being said the authors do not go further than this in their characterisation of the KO model, so the reviewer is right our data and their data is complementary but with different characterisation; we have now cited and discussed the Phu et al paper in the first paragraph of the discussion.

3. The fact the authors have described a potential phenotype with age in the USP30 knockout mice which is beneficial is worthy of further studies which include mechanistic work. This would strengthen the case that those beneficial effects of USP30 k/o with age are related to the deubiquitinase function of USP30 and not some other mechanism. Have the authors investigated the USP30 k/o / mitoKeima mouse during ageing in those tissues specifically protected from age-related disease?

We thank the reviewer for the comment on the possible ageing connection. Firstly, we wanted to make sure USP30 KO with age doesn't result in increased tumorigenesis. Fortunately, USP30 KO did not have any tumour predisposition and if anything aged better than WT mice. We don't know if this reflects better health with age (i.e. less accumulation of fatty liver) or an influence on the ageing capacity. We did cross USP30 mice to HGPS and POLG premature ageing mice to see if there is any rescue but USP30 KO/HGPS or USP30 KO/POLG double mutants were indistinguishable from HGPS alone or POLG alone. These suggested to us that USP30 influences health with ageing rather than ageing itself. We didn't add these data as considerable additional work, including greater animal number, is needed to adequately study this issue, and this is outside the scope of the current paper that focuses on USP30 and alpha-synuclein toxicity.

4. The authors validate work published by others that USP30 k/o increases mitophagy rates and have specifically investigated this in SN neurons, although the most relevant cell type to Parkinson's Disease, from a therapeutic perspective, it would be very beneficial for the authors to have investigated mitophagy rates in other tissues and even cell types within the brain. It is important to understand if the USP30 k/o effect is also present in glial cells of the SN and if the whole body effects are due to the mechanistic changes in mitophagy rates.

This is an important point; we have now quantified mitophagy in other parts of the brain (cortex, hippocampus) and other peripheral tissues such as the muscle and these data are now presented in Figure S3a and show now that USP30 KO mice show no difference in mitophagy in the other tissues. We also quantified mitophagy in other parts of the brain (cortex and hippocampus) and show that *Usp30* KO mice have higher mitophagy levels in these areas too (Suppl. Fig. 3 c,d). This is consistent with data generated ex vivo in *Usp30*KO hippocampal neurons published by Phu et al 2020.

We have also tried to quantify the mitophagy in the glia in the brain, but we have problems identifying if the mitophagosome is in the glia or in the surrounding tissue. For the benefit of the reviewer, we are providing some representative images in the figure below; as the reviewer might appreciate, it is hard to quantify if the red puncta is actually in the astrocyte in this case or in the surrounding neurons; we think we see more red-puncta, but because

we cannot be 100% sure we would rather not present this data at this point unless we could do these experiments also in some other models such as human mid-brain organoids, that is beyond of the purpose of this paper.

5. The study the authors carry out in the USP30 k/o animals with AAV-SNCA injection is very interesting and brings together multiple key pathways in Parkinson's; this is also novel for the study. Overall, the results are clear and well articulated.

We thank the reviewer for the positive comment regarding the importance of our AAV-SNCA study.

6. With respect to the TH neurons after AAV-SNCA injection, there appears to be a difference (quite large) between the effect this has in the WT mice versus the mitoKeima mice, however the authors have not commented on this. Could they please comment on this difference, why this could be and if this affects the resulting protective effects of USP30 k/o? Furthermore, the age at which the mice were injected with AAV-SNCA is of interest with the change in age related phenotypes, it would be interesting for the authors to comment on this. The authors show clearly the 2 genders in these studies, do the authors find any differences in response to AAV-SNCA in the USP30 k/o between genders?

Please see our response to reviewer 1, point #6 above for our response to this issue of the apparent difference in TH staining in WT versus mito-QC mice.

We have clearly indicated the age at which we did the injection and mentioned in the legend that we don't see any sex dependencies (no difference from males versus females). After we determined there are no differences, the animals were separated by sex in various experiments for ease of animal usage (i.e. females can be pooled in the same cage from different litters, males cannot leading to larger number of cages needed etc).

7. The authors then go on to validate their findings of the USP30 k/o with using a USP30 inhibitor and show the inhibition selectivity for this compound and in vivo characteristics. This is an important piece of validation work. The data shown on the inhibitor in Figure 4 is somewhat limited, at the very least it would be essential for the authors to add in detail on the robustness of the inhibition shown in Figure 4b. In addition, could the authors show the full panel of DUB and cathepsin inhibition rather than just 2 in Table 1?

We thank the reviewer for emphasizing how important the validation with a chemical inhibitor is. Regarding the potency of the USP30 inhibitor, it would be hard to add a panel as USP30 is uniquely selective to USP30 (no activity against other DUBs), but in the initial manuscript we did list the panel constituents in methods (initial Table 2, now Table 1).

8. Could the authors comment on why only female mice were used for the inhibitor study? In Figure 5c-e it appears as though the USP30 inhibitor treatment has reduced the dopamine, DOPAC and HVA levels in the EV treated mice. Could the authors comment on this and if they tested this statistically? In addition, did the authors test if the AAV-SNCA is significantly different to the AAV-SNCA + USP30 inhibitor group? The authors test 2 different doses of the inhibitor, however little dose dependent effect is seen with these doses,

could the authors comment on any dose dependent effect and if not, what type of study would be required to understand this?

We are sorry for the typo, we used male mice; we have corrected the statement now. Throughout our study, we haven't seen any differences in mitophagy or other parameters when we compared males to females. We chose though not to pull genotypes to keep interpretation clean.

Please see our response to reviewer 1, point #6 above, for our response to this issue of the apparent difference in TH staining in WT versus mito-QC mice.

9. The authors have shown blood levels of the inhibitor, however, the inhibitor treated dataset and study is hampered in usefulness as no target engagement work has been done, including mitophagy rates, ubiquitination assays etc. This makes the full interpretation of the USP30 inhibitor study difficult. Are there any differences between the effectiveness of the USP30 inhibitor in vivo in different cell types or tissues? These are crucial points to be addressed if the compound and target are to be progressed to the clinic.

We thank the reviewer for the critical suggestions. As mentioned in the response to Reviewer 1 we have now also included data on mitophagy in human neurons containing the Mito-QC tracker (Fig. 4 e,f) that show a dose dependent increase in mitophagy upon USP30i treatment in presence of A/O. Moreover, to show engagement, we have looked at the effect of the inhibitor on TOM20 ubiquitination in dopaminergic neurons derived from PD-patient derived iPS with the A53T mutation (Figure S6b,c) and show that in these cells too USP30i leads to increased TOM20-Ub. Other substrates were also validated, including SYNJ2BP, which is an interesting substrate given the potential role in mitochondria-ER contact sites – this was also elevated in A53T neurons compared to WT neurons. However, we did not include this data in the revised text as we believe this area of mito-ER contacts and Parkin/PINK1/USP30 biology should be studied further (see recent publication PMID: 37449534)

To further give information on the level of the inhibitor in the brain we have performed *CNS TE in WT animals after single dose (Figure S6d) showing that MTX325 engages the target at 50% level for approximately 8 hours after a 10mg/kg dose.*

10. In the discussion, the authors miss discussing their findings in relation to much of the current literature around USP30 and USP30 inhibitors, adding a fuller discussion of all of the USP30 inhibitor studies would greatly enhance the manuscript.

We have now added this to the discussion section.

11. Finally, the two study designs are different, in that the USP30 k/o study investigates protection of USP30 as the k/o is present before the AAV-SNCA pathology begins, whereas in the inhibitor experiments the AAV-SNCA is given first before inhibitor dosing begins. It would be useful if the authors could present data and/or comment on these two paradigms and what the results and effects could tell us about USP30 as a therapeutic target for PD.

We thank reviewer for the suggestion; we have added this to the discussion section.

12. A couple of minor points:

In the abstract there is a typo in line 3, it should read mitochondrial
Corrected

In the introduction line 4, the authors needs to add AR-PD
Corrected

The last line of page 3 has too many siRNA
Corrected.

The title of Supplementary figure 4 does not read correctly
Corrected.

Reviewer #3 (Remarks to the Author):

1. In this manuscript, Fang et al. describe a protective role for USP30 in counteracting the toxicity of a-synuclein (aSyn) in dopaminergic neurons. USP30 KO mice are protected in the AAV-aSyn overexpression model, and display reduced phospho-aSyn pathology and increased mitophagy in dopaminergic neurons. Pharmacological inhibition of USP30 produces similar benefits in mice. The significance of the mitophagy pathway for neurodegeneration in vivo is still debated and this manuscript builds on previous literature by showing that boosting mitophagy could provide protection against aSyn stress in vivo. Additional controls and experiments outlined below should strengthen the paper for publication in Nature Communications.

We thank the reviewer for the time and the very important suggestions to strengthen our paper.

2. Fig 1f - In the mitoQC images shown, the most noticeable change is an overall increase in the reporter signal (both red and green) as opposed to the appearance of "mCherry/red only" puncta. It will be better to highlight the mitophagy puncta counted in the images shown and provide zoomed-in images.

We are sorry for the confusion generated; we counterstained these images for LAMP1 to make sure we look at mitochondria and not peroxisomes (USP30 has been to also target peroxisomes); we have now added more explicit images for the quantification of mitophagy that include only the GFP and mCherry where no immunofluorescence for LAMP1 was performed.

3. The decrease in phospho-aSyn signal in USP30 KO (despite the preservation of dopaminergic neurons) mice is striking. The authors should provide more mechanistic insights into this observation. Is aSyn expressed to the same level between the genotypes (or is the total level of aSyn also reduced)? pSyn staining at an earlier time point should also inform if pSyn pathology forms normally but cleared faster in the KOs.

We thank the reviewer for the important note; we have looked more closely at the pattern of S129-SNCA phosphorylation and noticed that in WT AAV-A53T-SNCA injected mice the phosphorylated aSyn colocalises with the mitochondria (based on colocalisation with the mitochondrial marker OPA-1) such that about 80% of mitochondria are positive for S129-SNCA protein. In sharp contrast, colocalisation with mitochondria is much lower in the USP30 KO mice, such that only about 20% of mitochondria show colocalisation of S129-SNCA. While our data do not prove the mechanism of this observation, it is consistent with the possibility that increased mitophagy in the UPS30 KO mice leads to the preferential degradation of mitochondria that have S129-SNCA protein bound to the outer mitochondrial membrane, thereby resulting in fewer of the remaining mitochondria showing colocalisation with S129-SNCA. We present this data in Figure S5b,c,d. Unfortunately, to do the experiments at an early time point, this would take a considerable effort in both time and money that while providing granularity, and could not be possibly done in the frame time required to respond to this review.

We have also analysed total alpha-SNCA by western and found no significant differences between *Usp30* KO mice and controls. However, there is a high level of variability within the mice of the same group thus, we feel that the IHC results presented in Suppl. Fig.7c is more precise and definitive and also shows no significant difference. We didn't include the western because we feel it is not a definitive result, but we can include it as supplementary data if the reviewer or editor prefers it.

4. Is the baseline TH terminal intensity in the striatum different across groups in Figure 3A (+Ev image in Mito-QC/USP30 KO seems higher compared to the wild-type genotypes)?

Please see our response to reviewer 1, point #6 above for our response to this issue of apparent difference in TH staining in WT versus mito-QC mice.

5. The authors could discuss that intermittent dosing may be sufficient for efficacy given that the target coverage was between 7.5-12 hours daily.

Based upon updates to the modelling, we have slightly revised the text on estimated target coverage. As provided to previous reviewers' comment (reviewer 2 comment 9), lower doses of MTX115325 in a subsequent study did not show beneficial effects. It is difficult to speculate at this time whether high intermittent dosing would be sufficient for efficacy in this model.

6. Fig S4b - Around half of the AAV-aSyn injected animals did not develop the pSyn pathology even in the Vehicle dosed animals. What is the reason for this? Were injections QCed by following the needle track for proper access to the nigra?

We thank the reviewer for pointing this out. We noticed the same issue in the paper that we referred for generating the AAV-SNCA PD model (Ip et al, 2017) . To improve the efficiency of the PD model, we adjusted the coordinates for the stereotaxic injection and checked all mice at 28 weeks post-injection in our pilot study. We found all 15 mice in the AAV-SNCA group developed alpha-synuclein pathology. We used the same coordinates for the real experiments and did not exclude mice during the statistical analysis. It is also noteworthy that ALL the injections led to pathology for the Usp30 KO mice done in the academic labs. Similarly, in the inhibitor studies, it's notable that we found statistical significance DESPITE not excluding any of the mice. The impact likely would have been greater if mice that appeared to lack effective SN expression of SNCA were excluded but we avoided doing that to eliminate any potential bias. We also confirmed the alpha-synuclein pathology in all the mice (about half of the mice in each group) that are assigned for PFA fixation and IHC/IF experiments. We have checked, and also in the experiments done by the contract research organisations (CROs) the needles have been tracked. Furthermore, one cannot rule out that expression may be increased during the model, cause cell death which would lead to degradation of the signal?

REVIEWERS' COMMENTS

Reviewer #1 (Remarks to the Author):

This is important work and the manuscript has been greatly improved by revision.

The last sentence of page 5 still makes no sense to me as pexophagy would also result in colocalisation with LAMP1.

Reviewer #2 (Remarks to the Author):

Thank you to the authors who have addressed many of the reviewers comments.

A few things still remain for the authors to comment on/add.

The authors have added data from A53T iPSC derived neurons, however in the data in supp Fig 6 only data from A53T neurons is shown. Could the authors please add in comparison with WT neurons as well, this is important in understanding if USP30 inhibition simply has the effect at increasing mitophagy in all cells rather than cells which specifically have an impairment.

The authors did not address the point raised in the original reviews about the lack of dose dependency in the 2 doses tested in the animals.

With regards to the data from cells, the authors state "that MTX115325 drives mitochondrial quality control processes in neurons in vitro with concentration-dependent effects in the presence and absence of exogenous stimuli." However, the only data presented from neurons is the iPSC derived data, the SHSY5Y data are not neurons, they are neuroblastoma, they have different levels of expression of parkin, different mitophagy rates to differentiated neurons etc. Therefore could the authors revise this statement.

All the USP30 inhibitor data in neurons is shown after prolonged treatment, are there any mitophagy effects after shorter treatments, could the authors comment on why long treatment times in vitro are required?

Still the authors do not show the effect of USP30 inhibition or k/o on alpha synuclein pathology is driven by mitophagy, therefore some tempering of the discussion statements would be worthwhile.

In the limitations section the authors have highlighted this but then referred to the evidence showing it might be the case, this should be rephrased so that the reader is clear that the direct link with the mitophagy mechanism has not been shown and indeed other mechanisms could be responsible (some of the proteomics datasets from other USP30 studies indicate other pathways which could affect synuclein pathology).

Reviewer #3 (Remarks to the Author):

The authors addressed the major points raised in the review. In addition, quantification in Sup. Fig. 7c should be sufficient without including the Western blots. The manuscript is suitable for publication in Nature Communications.

EDITOR COMMENTS

1. *In order to proceed to publication we will require all closely relevant previously published work is cited in the manuscript, as previously requested by the reviewers, including the papers raised by reviewer 1. Please revisit reviewer 1's previous requests, and cite the papers mentioned before resubmission."*

We thank the editor for the quick review and positive comments on our manuscript. We revisited the reviewer 1's previous requests, and have now added citations so that we have cited all the relevant papers mentioned in the latest version of our manuscript.

2. *At the same time we ask that you edit your manuscript to comply with our policies and formatting requirements and to maximise the accessibility and therefore the impact of your work."*

We have modified the manuscript so that we now follow the policies and formatting requirements in the latest version of our manuscript.

3. *If you wish, an interesting image (but not an illustration or schematic) for consideration as a Featured Image on the Nature Communications homepage. The file should be 1200x675 pixels in RGB format and should be uploaded as a Related Manuscript File with the title "featured image suggestion". In addition to our home page, we may also use this image (with credit) in other journal-specific promotional material. If your featured image is chosen you will need to complete a Licence to Publish form which will be sent to you via DocuSign at a later stage.*

We thank the editor for considering our manuscript as a Featured Paper.

REVIEWER COMMENTS

Reviewer #1 (Remarks to the Author):

1. *This is important work and the manuscript has been greatly improved by revision.*

We thank the reviewer for your comment. We appreciate your effort in helping us to improve our manuscript.

2. *The last sentence of page 5 still makes no sense to me as pexophagy would also result in colocalisation with LAMP1.*

We agreed with the reviewer's comments on pexophagy and LAMP1. We used the colocalisation of red mCherry puncta with LAMP1 as an alternative strategy for measuring mitophagy in brain sections where the endogenous signal of mCherry-GFP is not easily detected with confocal microscopy. The text has now been revised accordingly in the manuscript.

Reviewer #2 (Remarks to the Author):

1. Thank you to the authors who have addressed many of the reviewers comments. A few things still remain for the authors to comment on/add. The authors have added data from A53T iPSC derived neurons, however in the data in supp Fig 6 only data from A53T neurons is shown. Could the authors please add in comparison with WT neurons as well, this is important in understanding if USP30 inhibition simply has the effect at increasing mitophagy in all cells rather than cells which specifically have an impairment.

Thank you for the reviewer's suggestion! We modified the text accordingly on page 11, and now include the requested data in supplementary Figure S6c.

2. The authors did not address the point raised in the original reviews about the lack of dose dependency in the 2 doses tested in the animals.

Thank you for the reviewer's suggestion! We found evidence of possible dose dependent efficacy of MTX115325 in reserving dopamine/HVA/DOPAC levels in the ipsilateral striatum of PD mouse model, although these data do not definitely establish a dose-dependency. Further dose level/regimen studies would be required to understand a complete concentration-response relationship in this model. The manuscript is revised accordingly on page 13 with an appropriately qualified statement.

3. With regards to the data from cells, the authors state "that MTX115325 drives mitochondrial quality control processes in neurons in vitro with concentration-dependent effects in the presence and absence of exogenous stimuli." However, the only data presented from neurons is the iPSC derived data, the SHSY5Y data are not neurons, they are neuroblastoma, they have different levels of expression of parkin, different mitophagy rates to differentiated neurons etc. Therefore could the authors revise this statement.

We thank the reviewer's comment and have revised the statement accordingly on page 12.

4. All the USP30 inhibitor data in neurons is shown after prolonged treatment, are there any mitophagy effects after shorter treatments, could the authors comment on why long treatment times in vitro are required?

Thank you for the reviewer's question! We have now included these data in Supplementary Figure 6 c and d.

5. Still the authors do not show the effect of USP30 inhibition or k/o on alpha synuclein pathology is driven by mitophagy, therefore some tempering of the discussion statements would be worthwhile. In the limitations section the authors have highlighted this but then referred to the evidence showing it might be the case, this should be rephrased so that the reader is clear that the direct link with the mitophagy mechanism has not been shown and indeed other mechanisms

could be responsible (some of the proteomics datasets from other USP30 studies indicate other pathways which could affect synuclein pathology).

We thank the reviewer's suggestion on our statement and have revised the text accordingly on page 17-18.

Reviewer #3 (Remarks to the Author):

1. The authors addressed the major points raised in the review. In addition, quantification in Sup. Fig. 7c should be sufficient without including the Western blots. The manuscript is suitable for publication in Nature Communications.

We thank the reviewer's positive comments on our manuscript. Please see the revised Fig S7, with the Western blots removed as suggested.